# Preselection of potential target spaces based on partial information by the supplementary eye field
Osamu Yokoyama ⊠ & Yukio Nishimura

Before selecting a saccadic target, we often acquire partial information about the location of the forthcoming target and preselect a region of visual space even before the target becomes visible. To determine whether the supplementary eye field (SEF) represents information signifying the potential target space, we examined neuronal activity in the SEF of monkeys performing a behavioral task designed to isolate the process of visuospatial preselection under uncertainty from the process of selecting a specified location. Our data showed that the activity of SEF neurons represented information about the potential target space instructed by symbolic cues. Increased activity of visuospatially selective SEF neurons encoded the potential target space, which could be a mechanism facilitating subsequent selection of an appropriate target. Furthermore, electrical stimulation of the SEF during the preselection period disrupted subsequent target selection. These results demonstrate that the SEF contributes to the preselection of potential target spaces based on partial information.

It is often necessary in a situation where the exact location of a potential target is uncertain to preselect a region of visual space before the target actually appears. For example, a driver may be concerned about the sidewalk side of the road to see if any pedestrians suddenly run out into the street. If something then appears in the preselected region, they immediately locate and identify it and react accordingly by moving their eyes to capture its trajectory or by moving parts of their body[1]. On the other hand, if something appears outside the preselected region, it can be appropriately ignored[2]. Such visuospatial preselection in uncertain conditions can be based on cues in the environment or cues given by other individuals, the meaning of which could be learned through past experiences. However, little is known about how and which brain regions are involved in such a selection of an expanse of visual space under uncertainty.

Visuospatial selection based on a symbolic cue involves at least two processes: retrieval of learned information about the cue and selection of space according to the retrieved information. Therefore, brain areas involved in visuospatial preselection based on the cue are likely to exist at the intersection of an input-evaluation system, which represents the learned association between the cue and the required behavioral response, and a response-construction system, which represents the target space and constrains the motor outputs to be executed. Here, we hypothesized that the supplementary eye field (SEF), which plays multiple roles in the executive control of eye movements[3], may be involved in spatial preselection based on partial information for the following reasons. Anatomically, the SEF of monkeys has ample connections with the prefrontal cortex[4], which

represents various information about external stimuli and their behavioral significance[5], and reciprocal connections with the frontal eye field (FEF)[6,7], which plays a functional role near the output stage for the execution of eye movement[8]; it also sends modulatory signals to occipital visual areas[9–11]. Reflecting these anatomical connections, there are neurons that respond to visual stimuli, neurons that increase the firing rate around the time of saccade, and neurons that exhibit both visual-related activity and saccade-related activity in the SEF[12]. Taken together, these findings suggest that the SEF is functionally positioned at an intersection between input-evaluation and response-construction functions. Indeed, SEF neurons change their activity during the course of learning arbitrary associations between visual stimuli and saccade responses[13]. Furthermore, SEF neurons show anticipatory, target-selection-related activity earlier than the FEF and intraparietal cortex does[14], suggesting that the SEF is upstream of other brain regions involved in spatial selection and thus plays a higher-order role in top-down spatial preselection.

The present study aimed to elucidate whether the SEF is involved in the preselection of a region in visual space. Thus, we investigated neuronal activity in the SEF of monkeys performing a behavioral task designed to specifically elicit the process of visuospatial preselection under uncertainty about a target location. We found that the activity of SEF neurons represented information about the potential target space according to partial information provided by a foveal, symbolic cue. Neurons showing visual-related activity exhibited a higher firing rate when the preferred direction was within the preselected region than when it was not, which could be a

Neural Prosthetics Project, Tokyo Metropolitan Institute of Medical Science, Setagaya, Japan. ⊠e-mail: yokoyama-os@igakuken.or.jp

mechanism facilitating subsequent target selection. In addition, electrical stimulation of the SEF during the preselection period altered subsequent target selection. These results reveal a neuronal mechanism in the SEF underlying preselection of a region of visual space based on partial information and the causal role of the SEF in the spatial selection of a potential target region under uncertainty.

## Results

### Monkeys used partial information to select appropriate targets

To examine whether the SEF represents partial information to help monkeys select a correct target, we trained two monkeys on a newly devised, saccadic choice task (Fig. 1a). In the task, while the monkeys fixated on a central fixation point, a colored shape (symbolic cue) was presented on the fixation point; the shape cued the animals to make a saccade to a target in the left or right visual hemifield in the subsequent part of the trial to obtain a reward. Two distinct colored shapes were used as cues for the right and left hemifield such that we could judge whether the recorded neuronal activity differed depending on the visual properties or the significance of the symbolic cue. After the symbolic cue disappeared and a brief delay period, a

choice target (small rectangle, 1° visual angle) was presented in both the left and right visual hemifields at various locations on the circumference of an invisible circle with a radius of 10° centered on the central fixation point. In most trials (75%), the placement of the two choice targets was left-up (LU) and right-down (RD), left (L) and right (R), or left-down (LD) and right-up (RU) (25% each); other configurations were used in the remaining 25% of trials. From the symbolic cue onset to the appearance of the choice targets ("preselection period"), the monkeys knew which side the correct choice target would be presented on but could not predict its precise location. Thus, during the preselection period, they could not prepare a specific motor command but needed to preselect the instructed hemifield and maintain this information until the choice targets appeared. Only after the choice targets were presented ("target determination period") could they identify the correct target and prepare a motor command to make a saccade to the location while withholding the execution of that saccade until the "go signal" (i.e., the disappearance of the fixation point).

During recording experiments, the monkeys made correct choices in >97% of all trials (mean [SD] correct choice rate over sessions: monkey A, 99% [1.5], 26 sessions; monkey P, 97% [3.3], 23 sessions). Even though the

**Fig. 1 | Behavioral tasks and recording sites in the SEF. a** Temporal sequence of behavioral events in the saccadic choice task. The appearance of visual cues (symbolic cues) that instructed the monkey to select a target in the L or R visual hemifield and the main configurations of the choice targets are shown below the task sequence. **b** Temporal sequence of behavioral events in the visually guided delayed saccade task. **c** Task order in each recording session. The position of the inserted electrode was fixed throughout the session once the recording began. **d** Recording sites in the

SEF. Neuronal activity was recorded from the SEF in the left hemisphere of two monkeys. (Left and right) Filled black circles indicate the recording sites. Red squares indicate the sites where intracortical microstimulation evoked saccades. (bottom) Top view of the frontal cortex in the left hemisphere of monkey P. The gray rectangle indicates the recorded area. A anterior, L lateral, M medial, P posterior, AS arcuate sulcus, PS principal sulcus.

combination of the symbolic cue and angle of the choice targets was variable across trials, the correct choice rate was high in all trial types (range: monkey A, 94–100%; monkey P, 93–100%), suggesting that the monkeys selected whichever target was in the instructed visual hemifield, rather than making a selection based on mere individual associations (rote memory) between the given visual clues (symbolic cue and choice target location) and response (saccade toward a certain direction).

## SEF neurons represent potential and then specified target locations

We analyzed the activity of a total of 201 neurons, which were well isolated offline after the experiments, across the left SEFs of two monkeys (monkeys A and P, 110 and 91 neurons, respectively; Fig. 1d) while they performed the saccadic choice task (Fig. 1a). In addition, the activity of these neurons was also recorded while the animals performed a standard visually guided delayed saccade task (Fig. 1b), in which a target stimulus was presented at one of six positions and the animals were required to make a saccade to the target after the go signal; this task allowed us to examine the neurons' properties related to visuospatial processing, such as direction selectivity (Fig. 3b) and visual- or motor-related activity (Fig. 6a). The monkeys repeated a block of these tasks 1–3 times in each session (Fig. 1c).

In the following analyses, only the success trials (trials in which the correct target was chosen and the criteria for eye position were fulfilled) were included. The activity of an example neuron in the saccadic choice task is shown in Fig. 2a, b. During the preselection period (light blue-shaded period and the period between the light blue- and gray-shaded periods in Fig. 2a, b), the firing rate of the neuron increased dramatically in response to the symbolic cue when the animal was instructed to choose a choice target in the left hemifield (i.e., the hemifield ipsilateral to the recorded hemisphere, Fig. 2a and orange lines in Fig. 2b). Even after the symbolic cue disappeared (the period between the light blue- and gray-shaded periods in Fig. 2a, b), this increased firing rate was maintained. In contrast, when the animal was told to select a choice target in the right hemifield, the neuron maintained a low firing rate during the preselection period (Fig. 2a and purple lines in Fig. 2b).

During the target determination period (gray-shaded period in Fig. 2a, b), the activity of the example neuron in response to the onset of the choice targets showed further dependence on the locations of the choice targets. In the left-instructed trials, the neuron exhibited a transient increase in its firing rate with the presentation of LU-RD choice targets, i.e., the trials in which the saccade target was in the LU direction (Fig. 2a). In the left-instructed trials when the choice targets were presented at L–R or LD-RU positions, the firing rate decreased in response to the appearance of the choice targets (Fig. 2a). In the right-instructed trials, the firing rate of the neuron was generally less than that in the left-instructed trials (Fig. 2a and purple lines in Fig. 2b), except for a transient increase when the choice targets were presented at LU-RD (Fig. 2a). These activity patterns indicated that this neuron sequentially encoded the potential target space and the specified target location as the task requirements changed. Regardless of the visual features (color and shape) of the symbolic cue (comparisons of L1 vs L2 and R1 vs R2 in Fig. 2a), the neuronal response patterns to each symbolic cue (L and R) were almost identical, indicating that the neuron encoded behaviorally relevant spatial information that could be extracted from the symbolic cue but not its visual features. In the visually guided delayed saccade task, the same neuron showed a transient increase in firing immediately after a target was presented in the left visual hemifield (Fig. 2c, d); specifically, the increase in activity was maximal when a target was presented to the left of the fixation point.

We examined which task-related factors were represented by SEF neurons in a time-resolved manner (10 ms sliding windows with a step size of 10 ms for the inverse interspike interval (iISI)[14]). For statistical analysis, we employed a set of two-way analyses of variance (ANOVAs) and either categorized neuronal activity representation into one of five task-related factors or did not assign them a category according to whether the mean iISI in a given window of a given neuron differed depending on the visual

properties of the symbolic cue, the instructed hemifield, and the configuration of the choice targets (see "Methods" for the classification criteria; Fig. 3a). The five task-related factors were defined as follows: (i) "potential target space": the left or right visual hemifield as indicated by the symbolic cue (Fig. 3a, blue); (ii) "target position": the left or right visual hemifield as indicated by the symbolic cue and angle of the choice targets (Fig. 3a, red); (iii) "object and choice targets": the visual feature(s) (color/shape) of the symbolic cue and angle of the choice targets (Fig. 3a, magenta); (iv) "object": the visual feature(s) of the symbolic cue (Fig. 3a, green); and (v) "choice targets": the angle of the choice targets (Fig. 3a, gray). Ultimately, this analysis indicated that the previously described example neuron represented the potential target space during the preselection period and the target position during the target determination period (Fig. 2b, horizontal color bar).

Task-related factor analysis of all analyzed SEF neurons revealed a striking pattern (Fig. 3a, left): 60% ($n = 121/201$, monkey A: 49% [$n = 54/110$], monkey P: 74% [$n = 67/91$]) of neurons represented the potential target space during the preselection period. During the target determination period, more than half of the neurons that had represented the potential target space (60% [$n = 73/121$], monkey A: 59% [$n = 32/54$], monkey P: 61% [$n = 41/67$]) also represented the target position before the go signal. Of all the neurons analyzed, 46% ($n = 92/201$, monkey A: 42% [$n = 46/110$], monkey P: 51% [$n = 46/91$]) represented the target position during the target determination period. Among these neurons, 79% ($n = 73/92$, monkey A: 70% [$n = 32/46$], monkey P: 89% [$n = 41/46$]) also represented potential target space during the preselection period, indicating that the majority of neurons that represented target position during the target determination period also represented potential target space in advance during the preselection period.

These characteristics were also reflected in the time course of the proportion of neurons that represented these factors (Fig. 3a, right). In response to the symbolic cue, approximately 20% of neurons rapidly (~300 ms) began to show activity representing the potential target space, and the proportion increased to 25% during the delay period. Once the choice targets were presented, up to 23% of neurons' activity represented the target position, whereas the proportion that represented the potential target space decreased rapidly to 7%. At most, 6% and 12% of neurons represented either the visual feature of the symbolic cue or choice targets, respectively, indicating that SEF neurons rarely discriminated the visual properties of the stimuli and did not reflect only the visual stimuli.

These results demonstrated that the majority of SEF neurons modulated their activity to represent information about potential target space even when the precise location of the saccade target was uncertain. Once the choice targets appeared, SEF neurons combined the information about the location of the choice targets with the information about the potential target space; this process might be involved in saccade target selection and results in activity representing target position.

We also examined whether the neurons representing the potential target space, i.e., the L and R visual hemifields, were spatially distributed in a biased manner in the SEF. Neurons representing the potential target space were found throughout most of the recorded sites, as were neurons preferring either the left or right hemifield (Supplementary Fig. 1). Left-preferring and right-preferring neurons were also often observed at a single recording site. These results indicate that there was neither clustering nor spatial dissociation of neurons representing the left and right visual hemifields in the SEF.

## SEF neurons visuospatially encode potential target space

We next examined how the representation of information about the potential target space is related to visuospatial selectivity by SEF neurons. Similar to neurons in other visual- and saccade-related areas, such as the FEF[15] and superior colliculus (SC)[16], SEF neurons show direction preference during visual-related and saccade-related activity[12]. We hypothesized that the way in which SEF neurons represent information about the potential target space is related to their direction preference in visual-related activity.

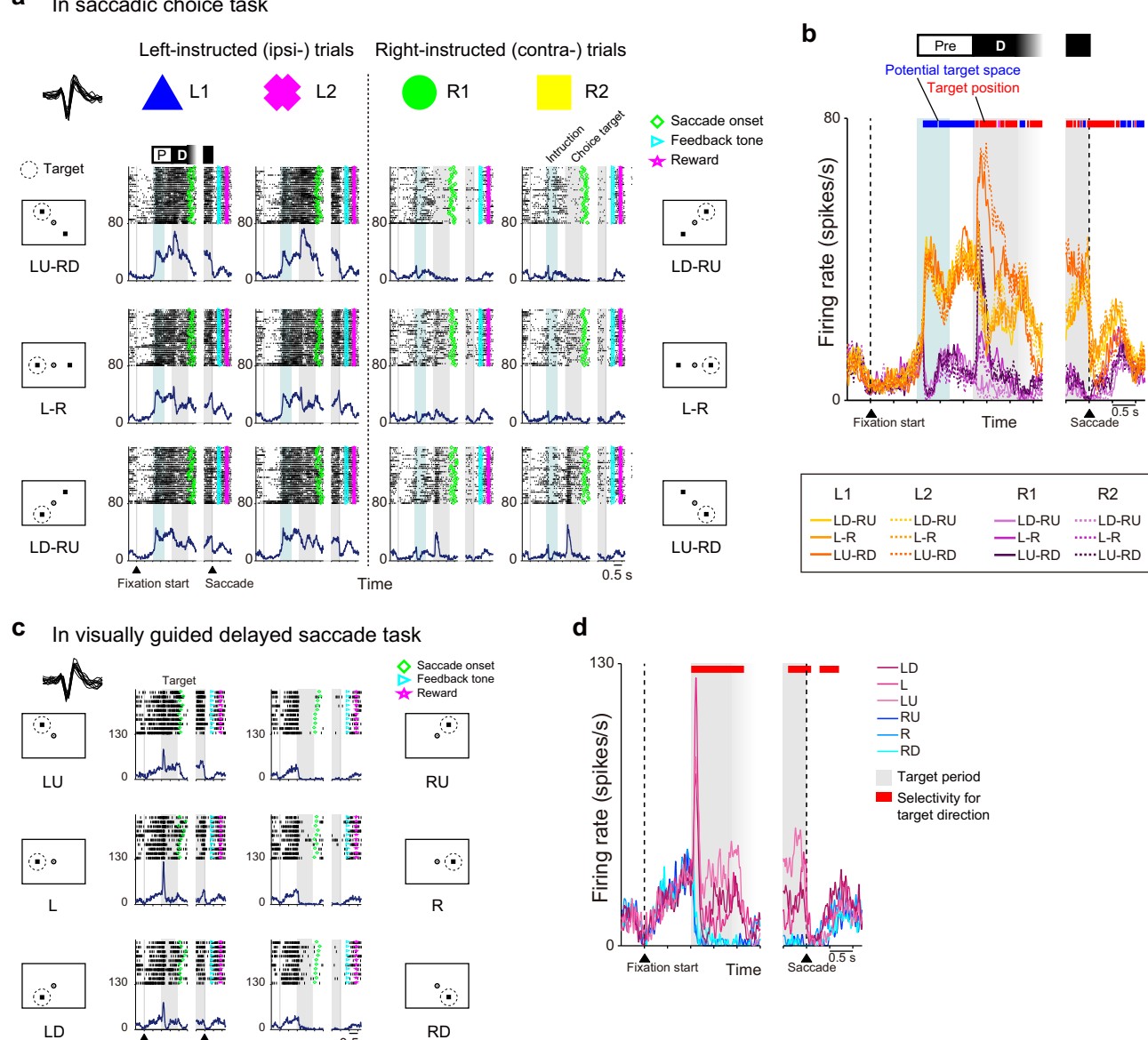

**Fig. 2 | Activity of an example neuron recorded from the SEF. a** Raster plots and mean spike density functions of an example neuron in the saccadic choice task. The ordinate represents the instantaneous firing rate (spikes s$^{-1}$). The plots are arranged such that trials with a certain symbolic cue are grouped in each column, and trials with a chosen target in a certain location are shown in the corresponding location of the panel. Neuronal activity is aligned with symbolic cue onset and saccade initiation (left and right gray vertical lines, respectively). Light blue shading indicates the period during which a symbolic cue was presented. Gray shading indicates the period from the choice target onset to the go signal (750–1200 ms). The spike waveforms are shown in the upper-left corner. **b** Merged spike density functions of the neuron shown in (**a**). The color of each line plot indicates the trial type, as shown below the plot. The color of the horizontal bar above the plot indicates the factors (blue: potential target space; red: target position) represented by the activity as statistically determined. **c** Raster plots and mean spike density functions of the same neuron are shown in (**a**) and (**b**) in the visually guided delayed saccade task. Gray shading indicates the period from the target onset to the go signal (750–1200 ms). Gray vertical bars indicate the fixation initiation and saccade onset times. The spike waveforms are shown in the upper-left corner. **d** Merged spike density functions of the neuron shown in (**c**). The color of each line plot indicates the trial type, as shown to the right of the plot. The red horizontal bar above the plot indicates the period during which the neuron exhibited directional selectivity.

To test this hypothesis, we first classified the recorded neurons into two groups, those with and without a visually direction-selective response to a target, according to activity in the visually guided delayed saccade task (Fig. 1b). Of the recorded neurons, 36% (*n* = 73/201) were classified as visually direction-selective (Fig. 3b), and their preferred directions (PDs) were identified (for the definition of PD, see Methods). Consistent with a previous study[12], the PDs of this subset of neurons were almost uniformly distributed across every tested direction, regardless of whether they were contralateral or ipsilateral to the recorded hemisphere (Fig. 3b, right; Rayleigh test, *p* = 0.9600 across the six directions tested).

In the saccadic choice task, the activity of neurons with visual direction selectivity accounted for 43% of the represented information about potential target space during the preselection period.

We then examined the relationship between each neuron's PD and its activity modulation in the saccadic choice task. Hereafter, we collapsed neuronal activity across the left- (or right-) instructed trials with different symbolic cues since the activity of SEF neurons rarely discriminated visual properties, as shown above (Fig. 3a). For specified target location, we confirmed that the SEF neurons exhibited greater activity when the chosen target was in the same direction as their PD than when it was not during the

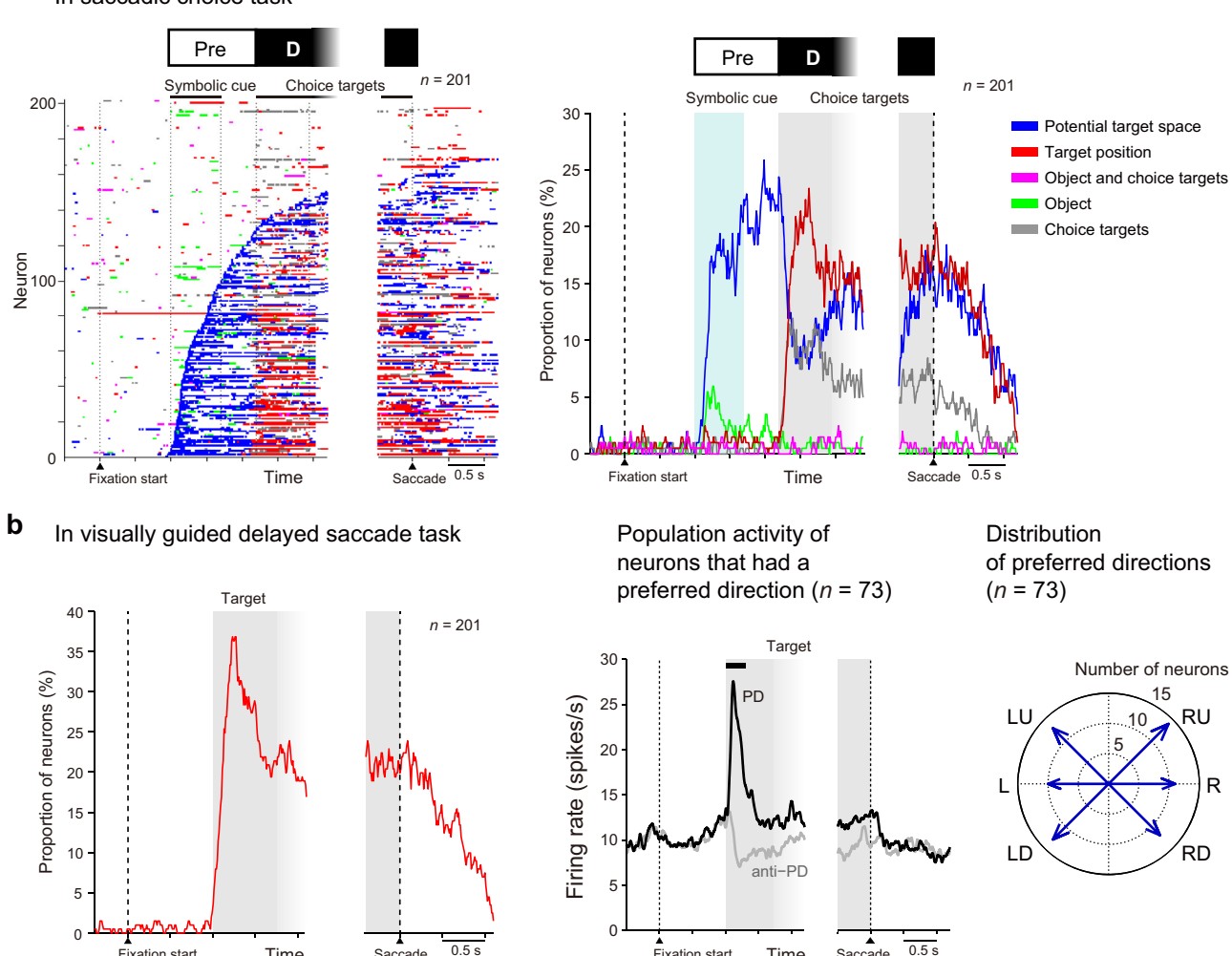

**Fig. 3 | SEF neurons represent the potential target space and target position.**
**a** Left, time courses of the representation of task-related factors by individual SEF neurons in the saccadic choice task. Different colors indicate selectivity for potential target space (blue), target position (red), object (green), choice targets (gray), and object and choice targets (magenta). Each row represents a single neuron. Neurons were ordered according to the onset time of selectivity for the potential target space. Right, the proportion of neurons representing task-related factors over time. The color code is the same as in the left panel. **b** Left, the proportion of neurons that exhibited direction selectivity over time in the visually guided delayed saccade task. Center, mean activity of neurons that had direction selectivity. The PD of each neuron was determined based on the spike counts during the 0–300 ms period after target onset (horizontal black bar above the plot). Population-averaged activity was calculated separately for the PD (black) and anti-PD (gray) trials. Right, distribution of the PDs of the neurons.

target determination period (Supplementary Fig. 2). This result is similar to a previous finding[17] that SEF neurons show greater activity when a single stimulus is presented in the neuron's PD, but our results extend this finding and note that SEF neurons also exhibit greater activity when a stimulus is presented in the PD even when another stimulus is presented outside the PD. For potential target space, we compared neuronal activity when the PD was within the instructed hemifield (PD-included condition) vs when it was not within the instructed hemifield (PD-opposite condition) and found that neuronal activity was greater in the PD-included condition than in the PD-opposite condition (Fig. 4a, b). More neurons exhibited greater activity in the PD-included condition than in the PD-opposite condition as early as 100–300 ms after symbolic cue onset (Fig. 4c) and during the preselection period (Fig. 4d, binomial test, $p \leq 0.015$, $\alpha = 0.05$, false discovery rate [FDR]-corrected for multiple comparisons). The degree of separation of neuronal activity between the conditions was examined by receiver operating characteristic (ROC) analysis (Fig. 4e). For the neuronal population, the area under the curve (AUC) was significantly greater than that of shuffled data (Wilcoxon signed-rank test, $p \leq 0.01$, $\alpha = 0.05$, FDR-corrected) in the middle of the symbolic cue phase and the middle to late delay phase (Fig. 4e),

indicating that neuronal activity was greater in the PD-included condition than in the PD-opposite condition. These results demonstrate that SEF neurons show greater activity when a neuron's PD resides within the potential target space than when it does not.

The coherent way in which SEF neurons represented the potential target space and specified target location is summarized in Fig. 4f (for the example neuron) and Fig. 4g (for the neuron population). During the preselection period, neuronal activity diverged depending on the relationship between the PD and the potential target space; specifically, neurons showed greater activity when the potential target space contained their PD than when it did not. Then, during the target determination period, neuronal activity changed further so that activity was greatest when the direction of the chosen target matched the PD.

Since the SEF is known to play an important role in eye behavior, we scrutinized eye position during the preselection period of the saccadic choice task. If the eye position differed depending on the instructed hemifield, the observed differences in neuronal activity might be accounted for by the difference in eye position. Although the animals were required to fixate on the central fixation point, we detected small (±0.14° average peak deviation,

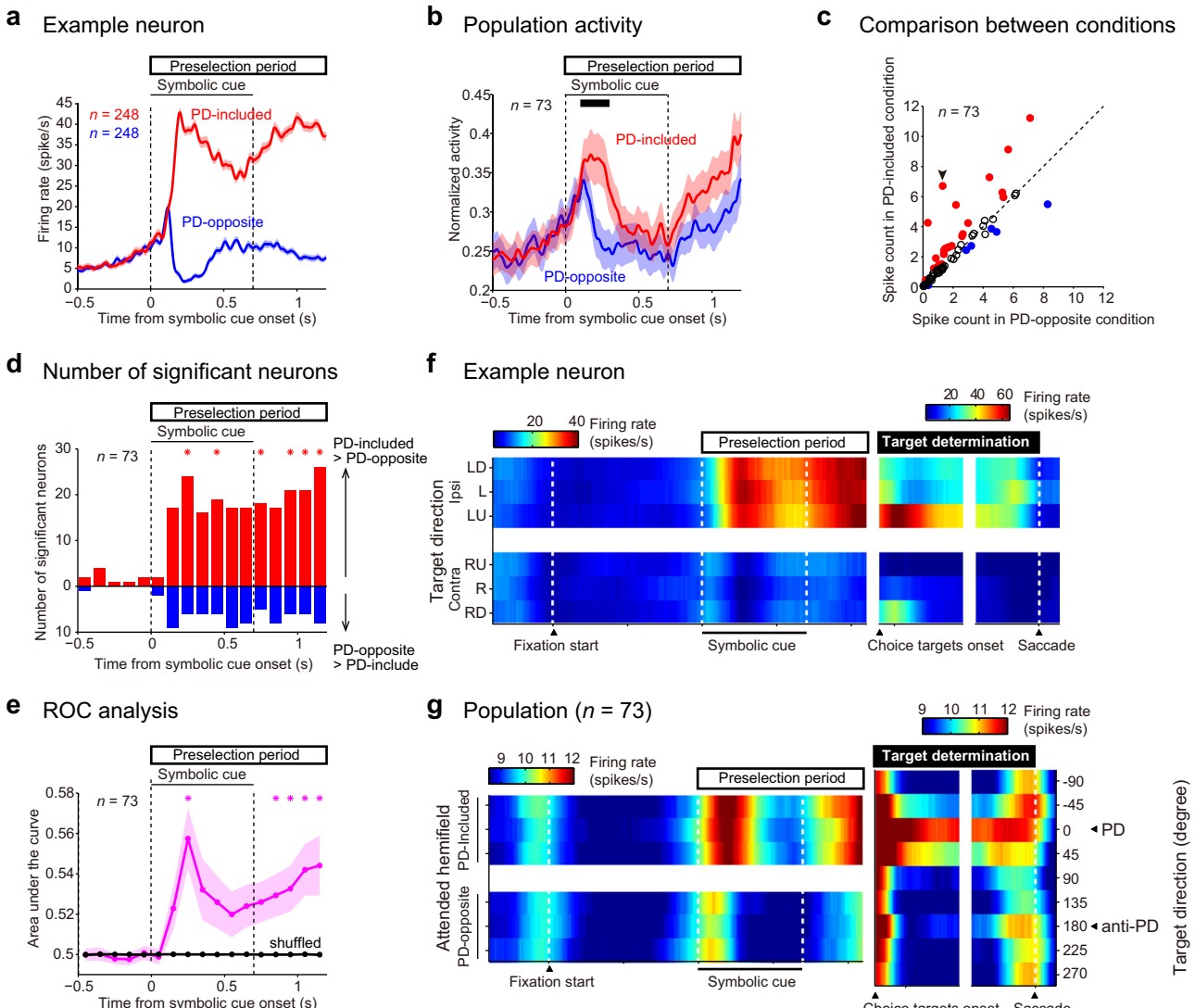

**Fig. 4 | SEF neurons encode the instructed visual hemifield if it includes their PDs. a** Activity of an example neuron in the PD-included and PD-opposite conditions (mean ± SEM) of the saccadic choice task. **b** Normalized activity for the population of neurons (mean ± SEM) for which a PD could be determined. **c** Comparison of spike counts at 0.1–0.3 s (black line in [**b**]) after symbolic cue onset. Red/blue-filled circles: neurons exhibiting greater/lesser activity in the PD-included than PD-opposite condition (two-sample *t*-test, α = 0.05), respectively. Black circles: neurons exhibiting no significance. Arrowhead: the example neuron in (**a**). **d** The number of neurons exhibiting different activity (two-sample *t*-test, *p* ≤ 0.0484, α = 0.05) between the PD-included and PD-opposite conditions. Asterisks indicate that the number of neurons showing significantly greater activity in the PD-included condition (red bar) was larger (binomial test, *p* ≤ 0.015, α = 0.05, FDR-corrected) than that showing greater activity in the PD-opposite condition (blue bar). **e** Time course of AUC values (mean ± SEM) for the PD-included vs PD-opposite condition across the neurons. Asterisks indicate that the AUC value of the experimental data

(magenta) was significantly greater (Wilcoxon signed-rank test, *p* ≤ 0.010, α = 0.05, FDR-corrected) than that of shuffled data (black). **f** Changes in spatial encoding by an example neuron. The mean firing rate is color-coded as a function of time (abscissa) and saccade target direction (ordinate). The color ranges are scaled separately for the periods before and after the choice target onset for illustrative purposes. **g** Changes in spatial encoding by a subpopulation of neurons for which a PD could be determined. The mean firing rate across the neurons is color-coded. For the period before choice target onset, the trial types (ordinate) were ordered so that the hemifield that included the PD and the hemifield that did not were aligned (shown on the left *y*-axis) across the neurons. For the period after the choice target onset, the trial types (ordinate) were ordered so that the PDs were aligned (to 0°, shown on the right *y*-axis) across the neurons. The color ranges are scaled separately for the periods before and after the choice target onset for illustrative purposes (shown by color scales above the plot).

*n* = 49 sessions across monkeys) but significant shifts in the relative horizontal eye position toward the instructed hemifield in response to symbolic cue presentation (Fig. 5a, b). This bias in eye position emerged ~0.17 s after symbolic cue onset (paired *t*-test, α = 0.01 for ≥20 ms; Fig. 5b). However, the shift in eye position was not consistent across all trials; in a relatively small proportion (11–42%) of trials, the eye position shifted to the hemifield opposite to the instructed hemifield (Fig. 5c, d, left). The results revealed that eye movements occurred even in response to partial information about the future target location and indicated that the monkeys extracted and processed the information conveyed by the symbolic cue.

Next, we examined whether the changes in neuronal activity during the preselection period could be accounted for by changes in eye position. Specifically, we compared neuronal activity between four groups of trials classified by the instructed hemifield (L/R) and the direction of the eye position shift (L/R). The results showed that the activity of an example neuron tended to be comparable between the trial groups with opposing eye position shifts when the instructed hemifield was held constant (L or R) (Fig. 5d, right). The results of a two-way ANOVA on the activity of all neurons indicated that there were many more neurons that showed a main effect of the instructed hemifield (blue in Fig. 5e) than neurons that showed a

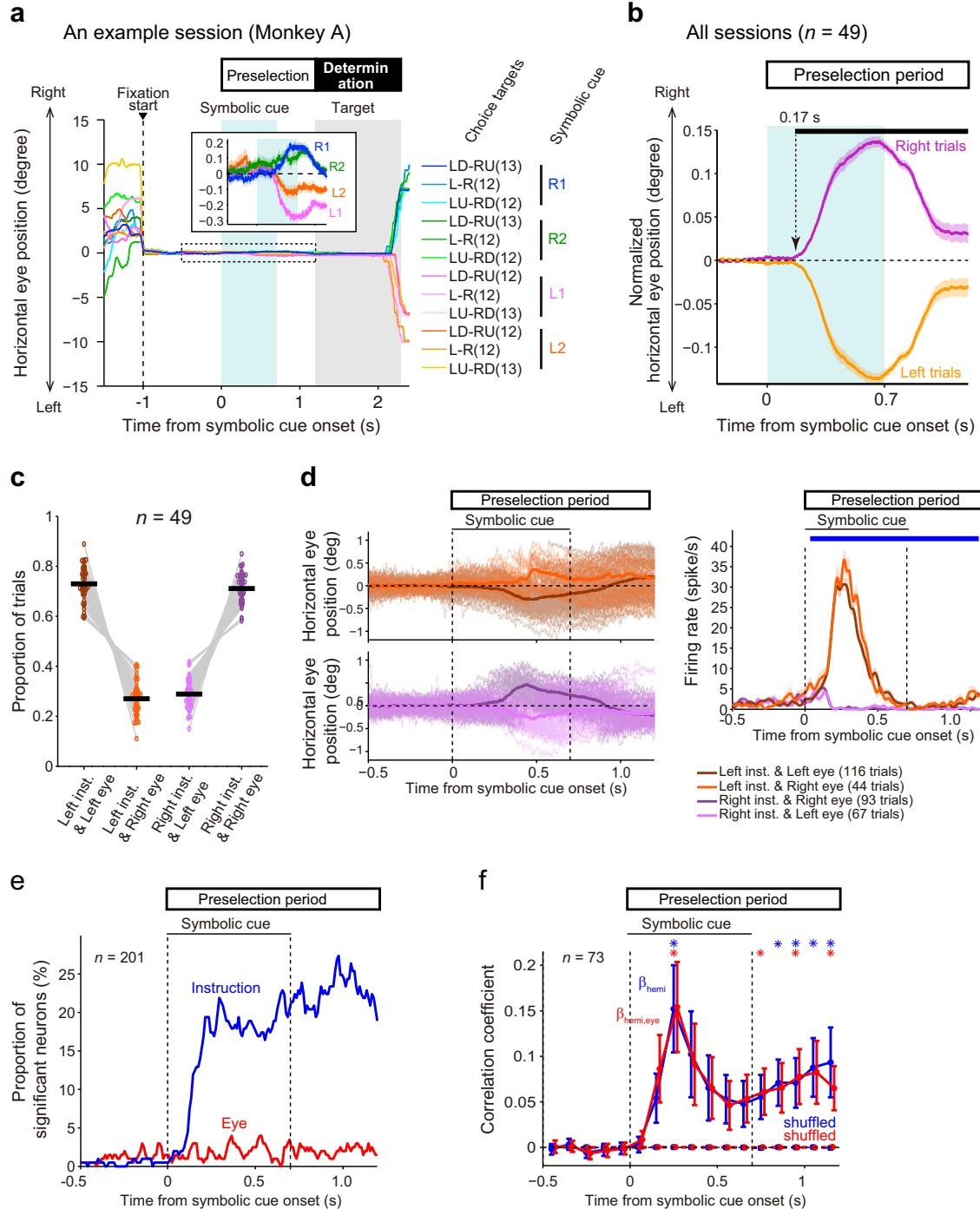

**Fig. 5 | Eye position bias and its effects on SEF activity. a** Horizontal eye position in the saccadic choice task of an example session. Each line indicates the mean for each trial type (i.e., a combination of instruction and configurations of choice targets). The color and number of trials for each trial type are shown on the right. The inset shows an enlarged view of the period marked by the dotted rectangle (from 0.5 s before the cue appearance until the choice target appearance) for trials grouped according to symbolic cue. **b** Normalized horizontal eye position (mean ± SEM across sessions and monkeys) during the preselection period. The black bar above the plot indicates the period during which eye position was different between the right- and left-instructed trials (paired *t*-test, α = 0.01 for ≥20 ms); the arrow indicates the onset time. **c** The proportion of the four types of trials classified by instructed visual hemifield (left/right) and the direction of eye position shift (left/right) across sessions. The black line indicates the mean. **d** Effects of instruction and eye position shift on neuronal activity. Left, eye position traces in individual trials (thin lines) and their mean (thick lines) for the four trial types classified by the

instructed hemifield (left/right) and the direction of the eye position shift (left/right) of an example session. Right, the activity of an example neuron in the four trial types (mean ± SEM). The blue line above the plot indicates the period in which the effect of the instructions was significant (two-way ANOVA, *p* < 0.01 in consecutive ≥3 bins). **e** The proportion of neurons that were modulated according to the instructed hemifield (blue) and that of neurons for which activity depended on the direction of eye position shift (red). **f** Effect of eye position on neuronal activity. The blue line shows the mean correlation coefficient ($\beta_{hemi}$) obtained when the spike counts were regressed by the instructed hemifield across neurons (mean ± SEM). The red line shows the mean partial correlation coefficient ($\beta_{hemi,eye}$) of the instructed hemifield when spike counts were regressed by the instructed hemifield and horizontal eye position across neurons. Asterisks indicate that the $\beta_{hemi}$ (blue) and $\beta_{hemi,eye}$ (red) values were greater than those of shuffled data (Wilcoxon signed-rank sum test, *p* ≤ 0.013, α = 0.05, FDR-corrected).

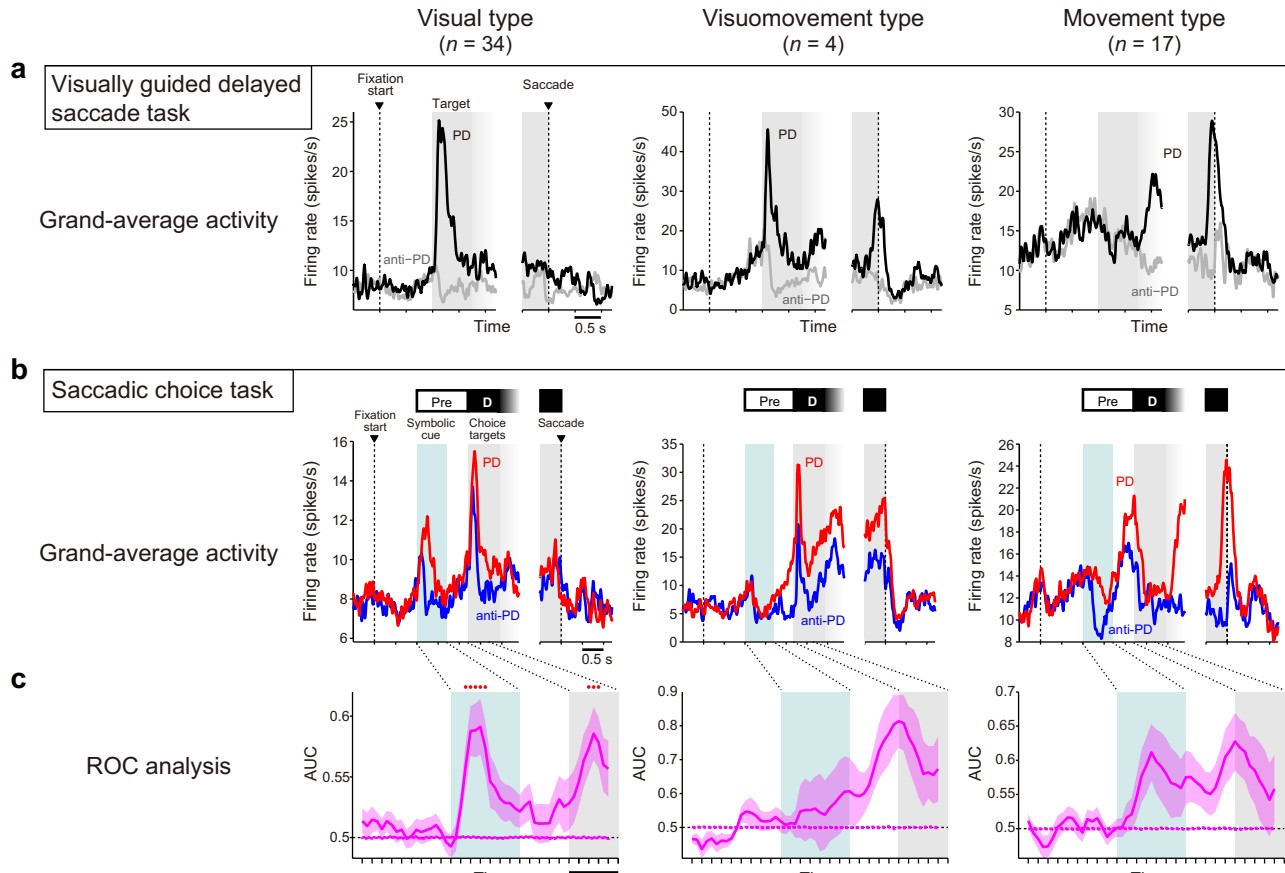

**Fig. 6 | Visual-type neurons encode potential target space. a–c** Left, visual type; center, visuomovement type; right, movement type. **a** Mean activity of the neurons in the PD (black) and anti-PD (gray) trials in the visually guided delayed saccade task. **b** Mean activity of the neurons in the PD (red) and anti-PD (blue) trials in the saccadic choice task. **c** Time course of the AUC values (mean ± SEM) for activity across the neurons in the PD vs anti-PD conditions. Red dots above the line plots indicate windows in which the AUC value of the experimental data (solid line) was significantly greater than that of shuffled data (dotted line) (Wilcoxon signed-rank test, α = 0.05, FDR-corrected).

main effect of the direction of eye position shift or the interaction between the instructed hemifield and the direction of eye position (red in Fig. 5e) throughout the preselection period. Furthermore, to examine whether the difference in neuronal activity between the PD-included and PD-opposite conditions could be accounted for by the differences in eye position during this period, single regression (with the PD-included/opposite condition as a regressor) and multiple regression (with the condition and horizontal eye position as regressors) analyses were performed. The partial correlation coefficient for the condition did not differ (paired $t$-test, $t$ (72) = 0.014–2.7, $p \geq 0.0080$, α = 0.05, FDR-corrected) based on whether eye position was included as a regressor (red in Fig. 5f) or not (blue in Fig. 5f). These results indicate that the majority of the difference in neuronal activity during the preselection period between the conditions reflect difference in the instructed hemifield but not differences in eye position.

These results demonstrate that spatially selective SEF neurons exhibited greater activity when their PD was located within the potential target space than when it was not, even when there was no visible object in the space and even when the monkeys maintained their gaze on the central fixation point. The selective increase in the activity of neurons with PDs in the potential target space could contribute to the detection and selection of the target appearing in the space as opposed to distractors appearing outside the space.

**Visually responsive neurons mainly encode potential target space**

On the basis of the activity pattern observed in the visually guided delayed saccade task (Fig. 1b), SEF neurons can be characterized as having a visual and/ or movement component[18]. We identified three types of neurons, i.e., visual,

visuomovement, and movement, and examined which type encoded the potential target space (Fig. 6). Neurons that increased their activity in response to visual stimulation (spike number$_{[0–300\ ms\ after\ target\ onset]}$ > spike number$_{[0–300\ ms\ before\ target\ onset]}$, α = 0.05 by Wilcoxon signed-rank test), but not around saccades (spike number $_{[−100–50\ ms\ around\ saccade\ onset]}$ ≃ spike number$_{[200–350\ ms\ before\ saccade\ onset]}$, α = 0.05 by Wilcoxon signed-rank test), were classified as visual type ($n$ = 34/201 [17%]; Fig. 6a, left). Neurons that increased their activity in response to visual stimulation (spike count$_{[0–300\ ms\ after\ target\ onset]}$ > spike count$_{[0–300\ ms\ before\ target\ onset]}$, α = 0.05 by Wilcoxon signed-rank test) and around saccades (spike count $_{[−100–50\ ms\ around\ saccade\ onset]}$ > spike count$_{[200–350\ ms\ before\ saccade\ onset]}$, α = 0.05 by Wilcoxon signed-rank test) were classified as visuomovement type ($n$ = 4/201 [2%]; Fig. 6a, center). Neurons that increased their activity around saccades (spike count$_{[−100–50\ ms\ around\ saccade\ onset]}$ > spike count$_{[200–350\ ms\ before\ saccade\ onset]}$, α = 0.05 by Wilcoxon signed-rank test), but not in response to visual stimulation (spike count$_{[0–300\ ms\ after\ target\ onset]}$ ≃ spike count$_{[0–300\ ms\ before\ target\ onset]}$, α = 0.05 by Wilcoxon signed-rank test), were classified as movement type ($n$ = 17/201 [8%]; Fig. 6a, right). The activity of each type was compared between the PD and anti-PD conditions in the saccadic choice task (Fig. 6b).

To quantify the degree of difference in activity between the conditions, ROC analysis was applied to the activity of individual neurons. The mean ROC values across neurons (Fig. 6c) revealed that visual-type SEF neurons exhibited greater activity during the preselection period, as well as during the target determination period in the PD condition than in the anti-PD condition (two-sample $t$-test between the experimental and shuffled data,

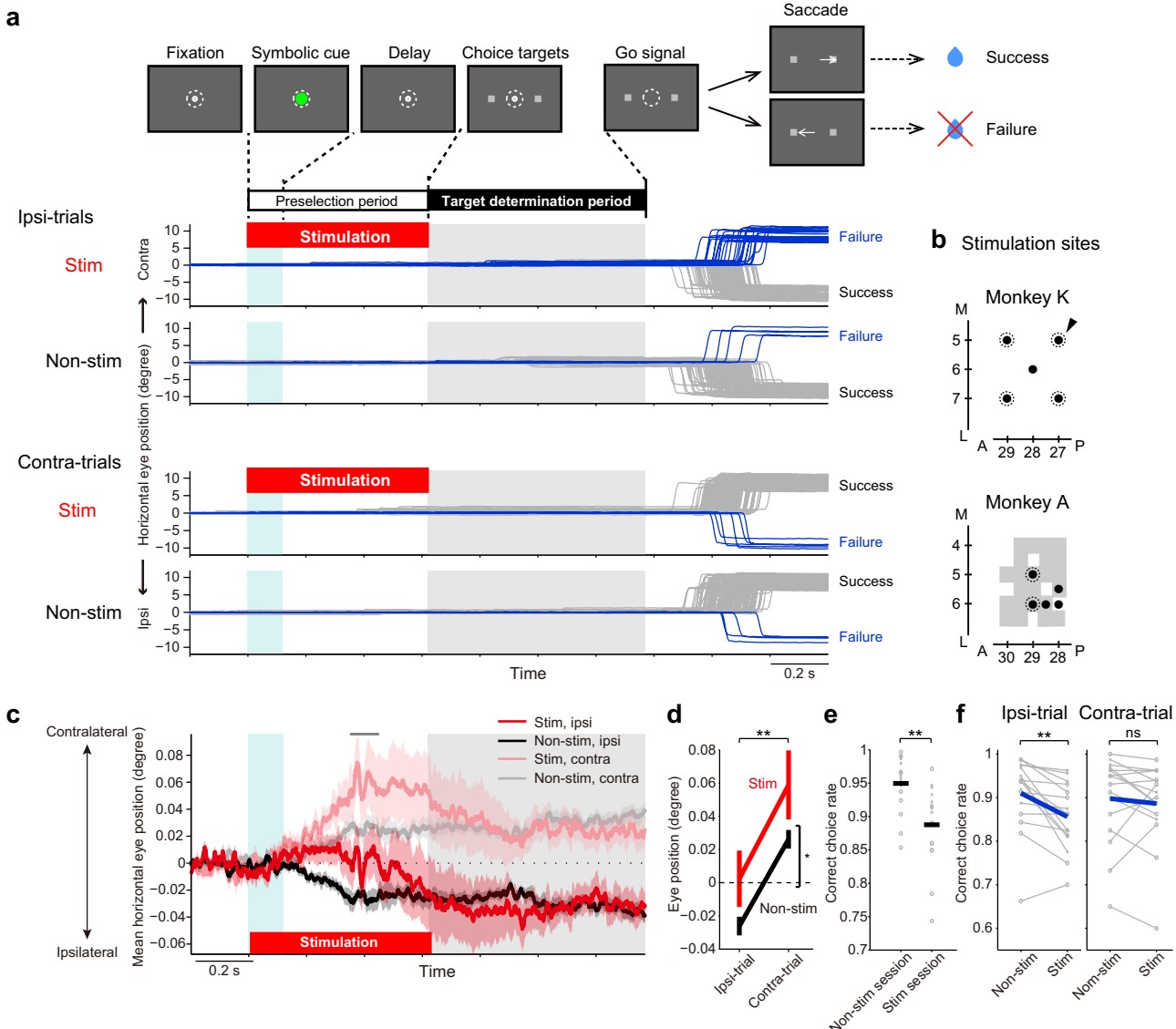

**Fig. 7 | Electrical stimulation of the SEF affects choice behavior based on spatial selection under uncertainty. a** Experimental procedure and horizontal eye position in an example session. Gray and blue lines indicate eye traces in the success and failure trials, respectively. Positive and negative values indicate the contralateral (right) and ipsilateral (left) sides, respectively. Ipsi- and Contra-trials indicate trials with the correct target in the visual field ipsilateral and contralateral to the stimulated (left) hemisphere, respectively. Stim, stimulation trials; non-stim, nonstimulation trials. **b** Stimulation sites are indicated with black-filled circles and the sites where intracortical microstimulation evoked saccades during free viewing outside of the task are encircled with a dotted line. Gray shading indicates the sites where neuronal activity was recorded. The arrow indicates the stimulation site in the example session shown in (**a**). **c** Horizontal eye position in two monkeys (mean ± SEM, n = 16). The scale of the ordinate was magnified from (**a**). **d** Horizontal eye position at 0.35–0.45 s (gray bar in [**c**]) after symbolic cue onset (and stimulation onset) in the two monkeys (mean ± SEM, n = 16; monkey A, n = 9 sessions; monkey K, n = 7 sessions). The horizontal eye position was more contralateral in trials with SEF stimulation than in those without SEF stimulation (two-way ANOVA, *p = 0.032) and more contralateral in Contra-trials than in Ipsi-trials (**p = 0.00026). **e** Effect of stimulation on correct choice rate (nonstimulation and stimulation sessions, n = 16, each; monkey A, n = 9 [circles]; monkey K, n = 7 [Xs]; two-sample t-test, **p = 0.0022). The black bar indicates the mean values; the light gray mark indicates individual sessions. **f** Correct choice rate in stimulation sessions (n = 16; nonkey A, n = 9 [circles]; nonkey K, n = 7 [Xs]). Blue indicates mean values; light gray indicates individual sessions. For the Ipsi-trials (left), the correct choice rate was higher in the Stim trials than in the non-stim trials (paired t-test, **p = 0.0024), whereas no difference was observed between the Stim and non-stim trials for the Contra-trials (right, p = 0.4625).

α = 0.05, FDR-corrected). In contrast, the activity difference of movement- and visuomovement-type neurons did not reach significance. These results indicate that neurons involved in visual processing play a leading role in the encoding of potential target space.

**Electrical stimulation of the SEF biases the preselection of space**

The above analyses demonstrated that the majority of SEF neurons encode potential target space; however, it remained unclear whether the SEF contributes to the preselection of potential target space. Therefore, we tested whether the activity of SEF neurons was causally involved in the preselection of potential target space by performing electrical stimulation experiments in two monkeys (monkeys A and K, 6 and 7 sessions, respectively). Electrical stimulation was applied to the SEF (Fig. 7b) during the preselection period of the saccadic choice task (Fig. 7a). We conducted trials with (stimulation trials) and without (nonstimulation trials) stimulation in a pseudorandom order within a session. We first confirmed that applying stimulation with the chosen intensity neither elicited saccades nor disrupted eye fixation during the preselection and target determination periods (Fig. 7a). However, close

inspection revealed that stimulation slightly affected fixational eye position (Fig. 7c). In the nonstimulation trials, consistent with the findings in the recording sessions (Fig. 5), the horizontal eye position shifted slightly toward the instructed hemifield (gray and black lines for ipsi- and contra-trials, respectively, in Fig. 7c). In the stimulation trials, the horizontal eye position gradually shifted to a position more contralateral to the stimulated hemisphere in response to stimulation during the preselection period, regardless of the instructed hemifield (red and pale red lines for ipsi- and contra-trials, respectively, in Fig. 7c), compared to the nonstimulation trials. Notably, in the ipsilateral trials (red line in Fig. 7c), SEF stimulation caused the eye position to shift to the contralateral side, contrary to the instructed direction of the potential target space (i.e., ipsilateral). This stimulus-induced shift in eye position in the ipsilateral trials approached that observed in the contralateral trials without stimulation (gray line in Fig. 7c). A two-way ANOVA revealed that at 0.35–0.45 s after symbolic cue onset (and stimulation onset), during which the divergence of eye position was most pronounced in the nonstimulated trials (gray vs black lines in Fig. 7c), the main effect of stimulation on eye position (Fig. 7d; $F_{(1,60)} = 4.80$, $p = 0.032$, $\eta^2 = 0.06$), as well as that of trial type (ipsi- vs control-trials, $F_{(1,60)} = 15.11$, $p = 0.00026$, $\eta^2 = 0.19$) were significant. In the subsequent response period, more erroneous choices were observed in stimulation sessions than in nonstimulation sessions (Fig. 7e; two-sample $t$-test, $t(30) = 3.36$, $p = 0.0022$, Cohen's $d = 1.12$), showing a disruptive effect of stimulation. In particular, more erroneous choices were observed in the ipsilateral trials with SEF stimulation compared to those without stimulation (Fig. 7f). This decrease in the correct choice rate in the ipsilateral trials with SEF stimulation ($-0.05$, mean $\pm$ SEM = $0.86 \pm 0.019$ in stimulation trials compared to $0.91 \pm 0.021$ in nonstimulation trials) was small, but statistically significant (Fig. 7f, left; $n = 16$, paired $t$-test, $t(15) = 3.65$, $p = 0.0024$, Cohen's $d = 0.91$). In contrast, in the contralateral trials, the correct choice rate was not statistically different between the stimulation and nonstimulation trials (Fig. 7f, right; $n = 16$, stimulation trials: $0.89 \pm 0.024$, nonstimulation trials: $0.90 \pm 0.025$; paired $t$-test, $t(15) = 0.75$, $p = 0.4625$, Cohen's $d = 0.19$). These results indicated that electrical stimulation of the SEF induced a bias of spatial selection toward the contralateral hemifield, which was reflected in the shift of eye position to the contralateral direction, resulting in a reduction in choice accuracy in the trials in which choosing an ipsilateral target was needed. These results demonstrate that the activity of SEF neurons is causally involved in the preselection of potential target space under uncertainty about the actual target position.

## Discussion

By using a task design that allowed us to isolate the process of visuospatial preselection under uncertainty from the process of selecting a specified location, our results demonstrated that SEF neurons represent information about a potential target space in response to foveal symbolic cues that convey partial information about the future target location. The increased firing rate of visually responsive neurons encoded the potential target space when their PD was included in the space. These activity changes were scarcely accounted for by eye position. In addition, electrical stimulation of the SEF during the preselection period influenced subsequent target selection. These results indicate that the SEF contributes to spatial preselection based on partial information about the target location, even in the absence of potential targets. These findings further extend our knowledge of the mechanism by which the brain selects visual space.

### Causal involvement of the SEF in spatial preselection

We demonstrated that the electrical stimulation of the SEF at subthreshold levels for inducing a saccade during the preselection period affected eye position and the subsequent selection of a saccade target (Fig. 7). This suggests that the SEF plays a causal role in the preselection of potential target space under uncertainty about the actual target position. The overall correspondence between the direction of very small stimulus-induced gaze shifts and erroneous spatial selection (Fig. 7c, d) is consistent with previous findings on the direction of microsaccades and covert attention[19,20]. The net

contralateral bias of the SEF to the contralateral visual field (Fig. 7c, d, f) is also consistent with previous findings on the direction of stimulation-evoked saccades[21,22], the endpoint of sequential saccades[23], and visual target detection and selection[24,25]. Our results add to these studies the contralateral bias of the SEF in selecting an expanse of space when the target position is uncertain.

There are several possibilities regarding the mechanism mediating the effect of electrical stimulation on eye position and subsequent target selection. First, the stimulation might induce discharges of SEF neurons involved in visual processing and consequently influence neuronal activity in occipital visual areas. Second, the stimulation might induce discharges of SEF neurons involved in eye movements. Third, the stimulation might induce discharges of SEF neurons that represented the conceptual information content of the instruction indicating the hemifield to be selected. Since the correspondence between the selected hemifield and the direction of the eye position shift was weak (Fig. 5c, d) and the population of SEF neurons representing eye position was much smaller than that of SEF neurons representing the instructed visual hemifield (Fig. 5e), the effect of electrical stimulation on target selection is unlikely to be mediated by neurons involved in eye movements. The significant but limited effect of SEF stimulation (Fig. 7f) might be due to the overtraining of the animals in the behavioral task. Another reason might be that other brain regions implicated in spatial selection, such as the FEF and parietal cortex, could compensate for functional deficits induced by the electrical stimulation of the SEF; notably, the SEF has reciprocal, anatomical connections[4,6,7] with these regions and they are thought to belong to the dorsal frontoparietal network[26,27] that controls spatial selection.

### Representation of an expanse of space by SEF neurons

Our data revealed that a subset of single neurons in the SEF successively represent information about both the potential target space and the direction of the chosen saccadic target (Figs. 2–4), suggesting that SEF neurons play a coherent role in spatial selection regardless of whether the target is a spatial region or a direction. The present results showed that visual direction-selective neurons exhibited a greater firing rate when their PDs were in the potential target space than when they were not (Fig. 4). When preselecting a region of space beyond that of the PD of each neuron, a subset of neurons that have various PDs in the potential target space exhibited an increase in their firing rate; this process could lead to the selection of a region of space by superposition of the neurons' PDs. The greater firing rate of these neurons before target appearance could then result in a greater firing rate in response to targets appearing in the preselected target space (a collection of PDs) compared to the rate elicited by targets appearing outside the potential target space (Fig. 4). We propose that this scenario is a plausible neurophysiological mechanism implemented by the SEF that underlies spatial preselection based on partial information, even before any target candidates appear. Because PDs differed across SEF neurons and were evenly distributed across all examined directions (Fig. 3b, right), this mechanism could result in the selection of any arbitrary region of the visual field. The lack of laterality in the PD of SEF neurons may be suitable for selecting a space spanning both the left and right visual hemifields, such as the upper or lower visual field. Future studies should investigate this possibility.

Our data (Fig. 2 for an example neuron) and those of previous studies[12,17] showed that SEF neurons exhibit broad direction-tuning curves, i.e., they respond to visual stimuli not only in their PD but also with lower firing rates to stimuli in directions adjacent to the PD. This property of SEF neurons may raise a concern about the unintentional selection of a space adjacent to the PD in addition to the PD. However, when the widths of the tuning curves (including PD and adjacent directions) and even distributions of PDs across the visual space (Fig. 3b, right) are accounted for, summation and superposition of tuning curves lead to a strong selection at the center of the targeted region and weaker selection of the region's periphery, with a diffuse boundary delineating the targeted and nontargeted space. Previous behavioral studies showed that the efficiency of attentional processing declines gradually, but not sharply, around the attended space[28,29]. The

inevitable consequence of the superposition of broad tuning curves of a population of neurons may be one of the reasons why the boundary between the selected and unselected space is not sharply demarcated.

## Possible roles of the spatial information encoded by SEF neurons

What is the role of SEF neuronal activity that encodes information about potential target space? How could this information be utilized in subsequent target selection? One possibility is that spatial encoding by SEF neurons is related to visuospatial attention. Mechanistically, as mentioned in the previous subsection, by increasing the discharge rate prior to the target's appearance, the visual response to a target candidate appearing in the preselected space reaches a certain firing rate more quickly than the visual response to another target candidate appearing outside the preselected space (Fig. 4). This facilitates the detection and selection of a target appearing in the preselected space but not outside it. This mechanism seems to be similar to that underlying visual attention in the visual brain areas[30]. Additionally, we observed that the gaze was slightly displaced toward the selected visual hemifield during the preselection period (Fig. 5b) and SEF stimulation (Fig. 7c). This finding is similar to previous findings that microsaccades occur in the direction of attention during covert attention[19,20]. Finally, mainly visual-type neurons represented the potential target space during the preselection period (Fig. 6). Collectively, the present results suggest that the role of SEF activity in this task seems to be more related to visuospatial attention. Indeed, previous studies of single units in monkeys[17] and human neuroimaging studies[31] have implicated the SEF in spatial attention when the target position is certain.

Another possibility is that spatial encoding by SEF neurons is related to saccade preparation. If neuronal activity leads directly to the neural process of the initiation of eye movement, it could be interpreted as being involved in saccade preparation. A previous study[32] proposed the following three criteria for considering a change in neural activity to be involved in motor preparation: (1) the change in neuronal activity must occur in the period preceding movement execution. (2) The activity change must reflect the information provided by the prior cue for the subsequent motor response. (3) The activity change must be predictive of some property of motor performance, such as reaction time. As for the first criterion, here, the activity change during the preselection period preceded the execution of the movement by definition (Fig. 1a). With regards to the second criterion, the activity change of SEF neurons reflected the information (i.e., which visual hemifield contained the correct saccade target) provided by the symbolic cue (Figs. 2–6). As for the third criterion, no relationship was found between SEF activity during the preselection period and eye movement metrics (latency from Go to the start of saccade, maximum velocity of saccade, and size of saccade) (Supplementary Fig. 3), indicating that motor performance could not be predicted from the activity of SEF neurons. Therefore, we cannot conclude that the preselection-related activity of SEF neurons obtained in the present study is involved in motor preparation.

However, since the task used in the present study required eye movements as a way of responding, the task design did not allow for dissociations of saccade preparation and visual attention and was thus unable to distinguish the role of SEF neurons. In future studies, it would be necessary to examine these possibilities by designing a task that dissociates saccade preparation and visuospatial attention (e.g., by dissociating the direction of saccade and the direction of attention or by using forelimb movements as a way of responding).

## Top-down roles of the SEF in spatial preselection

In our newly designed task, animals were required to select a possible target space according to a foveal cue, even before an actual target was present (Fig. 1a). In the preselection period, during which precise target location was uncertain, the activity of SEF neurons encoded a region of potential target space (Figs. 3–5), information that is essential to choose the correct target in the subsequent target determination period. Furthermore, the perturbation of SEF activity during the preselection period causally influenced subsequent choice (Fig. 7e, f). The results obtained from our task indicated that

the SEF plays a top-down role in spatial preselection under target location uncertainty. The top-down nature of the SEF has been documented by several previous findings. Neurons in the monkey SEF exhibit prolonged discharge (>300 ms) before spontaneous eye movements[22,33]. In a free-choice task[14], monkey SEF neurons with a given PD exhibited an anticipatory increase in their firing rate when the animals were aware that a forthcoming target would appear in that direction. Additionally, a brain imaging study showed that the SEF in humans exhibits increased BOLD signals while covertly attending to a peripheral location in the absence of visual stimulation[31]. It was also shown that the extent of activated areas in occipital visual regions increases as the extent of the attended space increases[34,35]; however, the neural mechanism controlling this activity has been elusive. Our results suggest that the SEF is the source of the signal-eliciting the activity increases in the downstream visual regions. Given that the SEF lacks direct anatomical projections to the occipital visual areas[4], other areas such as the FEF[9–11] may mediate the top-down influence from the SEF. Additionally, the influence of the SEF may be mediated by the SC because the SEF sends projections to the superficial layer of the SC, which also receives direct visual input from the retina[36].

## Nonspatial information encoding by SEF neurons

Our results showed that only approximately half (43%) of the information about potential target space during the preselection period was represented by the activity of neurons that exhibited visual direction selectivity in response to a peripheral target in a separate visually guided saccade task ($n = 73/201$, see Results). In other words, the rest (57%) of the information about the potential target space was represented by neurons without visual direction selectivity ($n = 128/201$). These direction-nonselective neurons might represent the visual hemifield in a conceptual and nonspatial manner, such as the meaning of 'R' or 'L', or the association between the symbolic cues and the visual space that was instructed to be selected based on learned association. Previous studies have shown that SEF neurons change their activity by learning new stimulus-response associations[13] and represent learned, nonspatial information associated with external, eye-behavior-related stimuli[37,38]. Anatomically, SEF neurons also have connections with the prefrontal cortex[4], which represents abstract information such as rules and categories in various cognitive tasks[39]; these connections imply that the SEF encodes some information about conceptual aspects of eye behavior, in addition to encoding motor-guiding signals related to eye behavior. These findings suggest that SEF neurons can represent spatial information not only in a spatial encoding space but also in an abstract, semantic space. The presence of information about potential target space represented both spatially and nonspatially in the SEF suggests that the conversion from conceptual, nonspatial information (meaning right or left, instructed by symbolic cues in the present study) to spatial information (selection of right or left visual hemifield in the present study) may also take place in the SEF.

## Methods
### Animals
Three Japanese macaque monkeys (*Macaca fuscata*), monkeys A, P, and K (two females, 5.4–5.6 kg, 4–5-years-old at the beginning of the experiment), were used. Neuronal activity recordings were conducted in monkeys A and P, and electrical stimulation experiments were conducted in monkeys A and K. They were cared for according to the National Institutes of Health guidelines and the Guidelines of the Tokyo Metropolitan Institute of Medical Science. All animal care and experimental procedures were approved by the Animal Care and Use Committee of the Tokyo Metropolitan Institute of Medical Science (approval nos. 13065, 14070, and 19049). We have complied with all relevant ethical regulations for animal use.

### Surgery
Before behavioral training, each monkey was implanted with plastic pipes to fix the head during the experimental sessions. Anesthesia was induced with ketamine hydrochloride (10 mg kg$^{-1}$ i.m.) with atropine sulfate, and aseptic

surgery was performed while anesthesia was maintained with pentobarbital sodium (20–25 mg kg$^{-1}$ i.v.). Antibiotics and analgesics were injected to prevent infection and pain, respectively. Polycarbonate and titanium screws were implanted in the skulls of the monkeys, and two plastic pipes were attached rigidly with acrylic resin. After the completion of behavioral task training, a second operation to permit access to the SEF was conducted under aseptic conditions. A part of the skull over the medial surface of the frontal lobe was removed, and a recording chamber was implanted using resin. The center of the chamber was placed at approximately +30 mm in the anterior-posterior direction.

## Behavioral tasks

The monkeys were trained on two tasks: a saccadic choice task (Fig. 1a) and a visually guided delayed saccade task (Fig. 1b). The tasks were run in blocks. During the experiments, each monkey sat in a primate chair with their head restrained and both forelimbs immobilized comfortably in plastic cylinders with straps. Eye position was monitored continuously at 240 Hz with an infrared video-based eye-tracking system (EYE-TRAC 6000 R-HS-S6; Applied Science Laboratories, Bedford, MA, USA). The tasks were controlled with the TEMPONET system (Reflective Computing, Olympia, WA, USA) with a precision of 1 ms.

## Saccadic choice task

In this task (Fig. 1a), the monkey was required to select either the right or left visual hemifield according to a symbolic cue, subsequently determine the saccade target presented within the hemifield, and select the target on the correct side with a saccade. At the beginning of each trial, a white circle (1° diameter, fixation point) was presented at the center of the black screen of a 27-inch LCD monitor (placed 59 cm in front of the monkey's eyes, ProLite B2776HDS-3; iiyama, Tokyo, Japan). When the monkey had gazed at the central fixation point for 1000 ms (fixation window size, 2.5° radius), one of four objects (symbolic cue; 2° × 2° visual angle) appeared on the fixation point. Each object cued the monkey to choose the target that would appear later in the trial within the left or right visual hemifield. For monkey A, an orange hexagon (L1) or purple hourglass (L2) cued choosing the target that would appear in the left hemifield, while a red diamond (R1) or cyan cross (R2) cued choosing the target that would appear in the right hemifield. For monkeys P and K, a blue triangle (L1) or pink X mark (L2) cued choosing the target that would appear in the left hemifield, while a green circle (R1) or yellow square (R2) cued them to choose the target that would appear in the right hemifield. The inclusion of four distinct objects was used to dissociate the effect of their cued information from that of their visual properties on neuronal activity. The symbolic cue disappeared 700 ms after its onset, and only the fixation point remained visible for 500 ms (delay period). Notably, from symbolic cue onset until the end of the delay period (preselection period), the animal knew which target in the left or right hemifield it should choose later in that trial but did not know the precise location of the target. Therefore, we assumed that during this period, the animal selected the left or right hemifield according to the symbolic cue. Subsequently, a pair of small gray squares (choice targets; each, 1° × 1° visual angle) was presented, one in each hemifield. The choice targets were positioned on the circumference of an imaginary circle (10° radius) centered on the fixation point. Across trials, they were presented at various symmetrical positions with respect to the central fixation point so that one was in the left hemifield and the other was in the right hemifield. The angle of the imaginary line connecting the two targets was variable across trials (from −45° to 45° with respect to the horizontal axis in steps of 5°). From the choice target onset to the go signal (the target determination period), the animal determined a choice target in the preselected hemifield as a saccadic target. After a variable wait period (750–900 ms for monkey A, 750–1200 ms for monkeys P and K), the fixation point disappeared (go signal), after which the monkey could make a saccade to the chosen target. If the monkey made a saccade toward the correct target within 1200 ms and maintained its gaze for 400 ms (hold period), a 500-ms high tone (600-Hz sine wave) was given as an indication of a successful trial. Then, a drop of liquid reward (~0.25 mL, apple juice for

monkey A, water for monkeys P and K) was delivered. If the monkey made a saccade toward an incorrect target and maintained its gaze for 400 ms, a 500-ms low tone (200-Hz sine wave) was given without reward delivery. After a variable (3600–4500 ms) intertrial interval (ITI), the next trial started. When the monkey's gaze left the fixation window before the go signal or left the target window during the hold period, the trial was terminated immediately without reward, and the next trial started after an ITI.

To ensure that the monkeys selected the entire left or right visual hemifield, it was essential to present the choice targets in unpredictable locations. To determine the presentation order of the symbolic cues and target locations, we used a randomized block design. Each block contained 12 regular trials and 4 additional trials. The regular trials consisted of the four symbolic cues and the three angles of the choice targets (−45° [left-up vs right-down], 0° [left vs right], and 45° [left-down vs right-up]), resulting in 12 trial types (4 symbolic cues × 3 angles). In the four additional trials, the choice targets were presented at angles different from those of the regular trials; the angles of the choice targets were chosen randomly from ±5–40° (in 5° steps). These additional trials were used to facilitate preselection of either the left or right visual hemifield even when the target position was still uncertain. Within each block, the order of trial types was determined pseudorandomly. If a monkey failed to perform a trial successfully, the same trial type was presented later at a pseudorandom time within the same block. As a rule, an unsuccessful trial was not repeated immediately to prevent the monkey from predicting a forthcoming trial type. When the monkey successfully performed every trial type in a block, a new trial sequence in the next block was determined.

After training for several months, the monkeys were able to choose the correct target at a high rate (>95%), and the recording experiments were started.

## Visually guided delayed saccade task

A conventional visually guided delayed saccade task (Fig. 1b) was used to determine the basic response properties of individual neurons, such as their preferred direction (PD, see the definition described in the subsection "Relation between PD and representation of the instructed hemifield" below) and whether their activity exhibited visual- or saccade-related modulations. Each trial commenced when a white circle (fixation point, 1° diameter) was presented at the center of the screen. If the monkey gazed at the fixation point for 1000 ms, a gray square (target stimulus, 1° × 1° visual angle) was presented at one of six positions (right-up [RU], right [R], right-down [RD], left-down [LD], left [L], and left-up [LU]) on an imaginary circle (10° radius) centered on the fixation point. After a delay period (750–900 ms for monkey A, 750–1200 ms for monkeys P and K), the fixation point disappeared (go signal), indicating that the monkey should make a saccade to the target. If the monkey made a saccade within 1200 ms and kept its gaze within the target window for 400 ms (hold period), a high tone (600 Hz sine wave, 500 ms) was given. Afterward, as a reward, a drop of liquid was delivered, and a variable (3600–4500 ms) ITI followed. When the monkey's gaze left the fixation window before the go signal or left the target window during the hold period, the trial was aborted, and an ITI began. The target position was pseudorandomized so that the monkey made a saccade to every target position in a block of six trials. Unsuccessful trials were presented again later randomly in the same block.

## Physiological recordings

To locate the SEF, the stereotaxic coordinates were used as an approximate guide (A25–30, L2–7), and intracortical microstimulation was applied (44 cathodal pulses of 200 μs at 333 Hz, ≤80 μA; adopted from[40,41]) through the tip of glass-coated tungsten electrodes (Alpha Omega, Alpharetta, GA, USA) inserted into the medial frontal cortex. On the basis of the outcome of intracortical microstimulation, the SEF was determined operationally as the sites where saccades were evoked and their vicinity (Fig. 1d).

After the SEF was located, all neuronal activity was recorded with a linear-array multi-contact electrode that had 16 contacts with a spacing of 150 μm (U-probe; Plexon, Dallas, TX, USA). The electrode was inserted into the brain through a 25-gauge guide tube that penetrated the dura mater

under the power of a hydraulic Microdrive (MO-972; Narishige, Tokyo, Japan) that moved the electrode in micrometer steps. The guide tube and electrode were inserted vertically (i.e., in the dorsoventral direction) into the cortex. Signals from the 16 contacts were amplified and bandpass filtered (from 500 Hz to 8 kHz). All waveforms that passed a threshold (3.5 SD) for amplitude were digitized (sampling rate: 24.414 kHz or 48.828 kHz; PZ2 and RZ2; Tucker-Davis Technologies, Alachua, FL, USA) and stored on a computer (Z220; Hewlett-Packard, Palo Alto, CA, USA).

In each recording session, the monkeys first performed 10 blocks of the saccadic choice task ($16 \times 10 = 160$ trials), followed by 10 blocks of the visually guided delayed saccade task ($6 \times 10 = 60$ trials), without any explicit instruction to inform them of task switching (Fig. 1c). The monkeys usually alternated between these tasks two or three times. The number of trials per neuron included in the analyses ranged from 200 to 667 trials (mean = 452 trials).

## Spike sorting

Single-unit potentials were isolated offline using a spike-sorting software package (OpenSorter; Tucker-Davis Technologies, Alachua, FL, USA). Drifting neuronal activity was excluded from all analyses.

## Eye position

The eye position time-series data during the saccadic choice task were segmented into individual trials. The mean value during a baseline period (0.5 s prior to symbolic cue onset) was subtracted from the data in a trial-by-trial manner, and the data were aligned with the timing of symbolic cue onset and averaged across trials for each condition (left- or right-instructed trials). Because we were interested in the effect of the symbolic cue, the averaged value for each condition was normalized by subtracting the overall mean value. To assess whether the eye position shifted to the left or right hemifield in each trial (Fig. 5c, d), we first determined the time point at which the difference in eye position between the left- and right-instructed trials was maximal for each session. Then, trials in which the eye position was in the left/right hemifield at the determined time point were defined as trials in which the eye position shifted to the left/right hemifield.

## Statistics and reproducibility

The data for neuronal activity and the effects of electrical stimulation were each collected on two monkeys. No statistical methods were used to pre-determine sample sizes, such as the number of animals, neurons, and trials, but they are comparable to previously published neurophysiological studies in nonhuman primates. All the statistical analyses were performed in Matlab R2013a (MathWorks, Natick, MA, USA). Statistical analysis of circular data was performed using CircStat[42]. We used two-tailed tests and significance was set at $\alpha = 0.05$ unless otherwise stated.

To examine which task-related factor was represented by neuronal activity (Fig. 3a), we performed two two-way ANOVAs step-by-step for sliding windows as described in the subsection "Representation of task-related factors" below. To determine whether the activity of a neuron had a PD (Fig. 3b), a one-way ANOVA was conducted as described in the subsection "Relation between PD and representation of the instructed hemifield" below. To test whether the PDs of the neurons were uniformly distributed across directions (Fig. 3b, right), we performed a Rayleigh test. To compare the activity of a neuron between the conditions during a certain period (e.g., Fig. 4c for 100–300 ms), the spike count was compared between the conditions using a two-sample $t$-test. To examine temporal modulation (Fig. 4d), the number of neurons that exhibited significantly greater activity in the PD-included condition compared to the PD-opposite condition and the number of neurons that exhibited significantly greater activity in the PD-opposite condition compared to the PD-included condition were compared using a binomial test ($\alpha = 0.05$, FDR-corrected) in each sliding window (duration, 100 ms; step size, 100 ms). To examine the time course of the degree of the difference in neuronal activity between the conditions (Figs. 4e and 6c), we performed ROC analysis[43] using sliding windows (duration, 100 ms; step size, 100 ms). An AUC value of 1 indicates that the spike count in the PD-included condition was always greater than that in the PD-

opposite condition, while a value of 0 indicates the opposite. An AUC value of 0.5 indicates that the spike count distribution was identical in the PD-included and PD-opposite conditions. The significance of the AUC value was assessed across neurons using a paired $t$-test ($\alpha = 0.05$, FDR-corrected) by comparing the value from the experimental data and the value obtained from shuffled data (the correspondence between neuronal activity and condition was shuffled, 1000 repeats). To examine whether eye position differed between the left- and right-instructed trials (Fig. 5b), paired $t$-tests were performed at every timepoint using averaged values for individual sessions across recording sessions ($\alpha = 0.01$ for $\geq$10-ms duration). To test whether changes in neural activity during the preselection period were related to the preselection of a potential target space or a small shift in eye position (Fig. 5e), we conducted two-way ANOVA with the instructed visual hemifield (left/right) and the direction of the eye position shift (left/right) as factors. Two-way ANOVAs were applied to the iISI in each sliding window (duration, 10 ms; step size, 10 ms). If the main effect of the instructed hemifield was significant ($p < 0.01$ in consecutive $\geq$3 bins), the bin was considered modulated by the instructed hemifield. If the main effect of the direction of the eye position shift or the interaction was significant ($p < 0.01$ in consecutive $\geq$3 bins), the bin was considered modulated by the eye position shift. Additionally, to examine whether the small shifts in eye position could account for the difference in neuronal activity between the PD-included and PD-opposite conditions (Fig. 5f), we performed single and multiple regression analyses for each neuron. The spike count in each sliding window (duration, 100 ms; step size, 100 ms) was regressed by the instructed hemifield (1 for PD-included, -1 for PD-opposite),

$$\text{Spike count} = \beta_{hemi} \times [\text{instructed hemifield}] + \varepsilon$$

or by the instructed hemifield (1 for PD-included, $-1$ for PD-opposite) and the horizontal coordinate of eye position,

$$\text{Spike count} = \beta_{hemi,eye} \times [\text{instructed hemifield}] + \beta_{eye} \times [\text{horizontal eye position}] + \varepsilon$$

For the neuronal population, the resultant correlation coefficients ($\beta_{hemi}$ or $\beta_{hemi,eye}$) were compared with those obtained from multiple regression analysis using shuffled data (1000 repeats) or with each other (paired $t$-test, $\alpha = 0.05$, FDR-corrected). To classify neurons into visual, visuomovement, movement, or none types (Fig. 6), we performed the Wilcoxon signed-rank test as described in detail in the subsection "Classification of neurons" below. To examine the effects of electrical stimulation and instructed hemifield (Fig. 7d), a two-way ANOVA was conducted. To compare the correct choice rate between the stimulation and nonstimulation sessions (Fig. 7e), a two-sample $t$-test was conducted. To compare the correct choice between the stimulation trials and nonstimulation trials (Fig. 7f), a paired $t$-test was conducted.

## Representation of task-related factors

To examine which task-related factor was represented by individual neuronal activity at each timepoint throughout the trial (Fig. 3a), we conducted a set of two-way ANOVAs for sliding windows (duration, 10 ms; step size, 10 ms). The beginning of the first window was aligned with the beginning of fixation on the central fixation point or the onset of saccade to the choice target across trials. The placement of previous sliding windows also allowed their alignment with symbolic cue onset and choice target onset. For each neuron, the inverse interspike interval (iISI)[14] throughout each trial, and the mean of the iISI in each window was calculated and subjected to the following statistical analysis. First, to examine whether neuronal activity reflected the visual features of the symbolic cue, for each pair of symbolic cues indicating the left or right hemifield (L1 and L2 or R1 and R2, respectively), we conducted a two-way ANOVA with an object (L1 or L2, or R1 or R2, respectively) and the angle of the choice targets ($-45°$, $0°$, or $45°$) as factors. If the main effect of the object was significant for at least one of the

pairs (α = 0.005 for each pair), we concluded that the neuron represented the object. If both of the main effects of object and angle were significant or the interaction between object and angle was significant (α = 0.005 for each pair), the neuron was considered to represent the object and the location of the choice targets. Next, only for neurons without selectivity for object or object and angle, we conducted another two-way ANOVA with the instructed visual hemifield (i.e., right or left) and the angle of the choice targets (−45°, 0°, and 45°) as factors. On the basis of the results of the second ANOVA (α = 0.01), we classified the neurons into four categories: (1) neurons that represented potential target space (only the main effect of the instructed hemifield was significant), (2) neurons that represented the location of the choice targets (only the main effect of the angle was significant), (3) neurons that represented target position (the main effects of the hemifield and angle were significant, or the interaction between the hemifield and angle was significant), and (4) neurons with no selectivity (not significant for any of the main effects and interactions). Overall, this analysis led to the classification of neuronal activity in each −10 ms window into six categories representing (1) the object, (2) the object and the location of the choice targets, (3) the potential target space, (4) the location of the choice targets, (5) the target position, and (6) none of the factors. In addition, we considered only the neuronal activity that represented the same category in at least three consecutive windows as representing that category.

### Relation between PD and representation of the instructed hemifield

Neuronal activity during the preselection period was analyzed. These analyses were performed on neurons with a PD. The PD of each neuron was determined based on the visual-related activity to target appearance (0–300 ms) in the visually guided delayed saccade task (Figs. 1b and 3). A one-way ANOVA was conducted to determine whether the spike count in this period was different across six different target positions. If this main effect was significant (α = 0.05) and if a certain direction maximally elicited spikes, the direction was defined as the PD. We found that 73 of 201 neurons (36%) fulfilled these criteria and included them in our analysis. For each neuron, the trials were divided into two groups. Trials in which the instructed hemifield included the neuron's PD were grouped into the PD-included condition; trials in which the instructed hemifield did not include the neuron's PD were grouped into the PD-opposite condition. To assess the effect of the instructed hemifield across the neuronal population (Fig. 4b), the activity of each neuron was normalized as follows. First, the minimum ($FR_{min}$) and maximum ($FR_{max}$) activity levels throughout the task (from 0.1 s prior to initial fixation to 1.2 s after choice target onset [around a saccade]) were determined across the PD-included and PD-opposite conditions. Next, the activity at each timepoint ($FR_t$) was normalized by calculating $(FR_t − FR_{min})/(FR_{max} − FR_{min})$ so that minimum activity became 0 and maximum activity became 1. Then, separately for the PD-included and PD-opposite conditions, the normalized activity was averaged across the neurons at each time point, resulting in normalized population activity (Fig. 4b).

### Temporally changing spatial representation

To visualize temporally changing spatial representation throughout the task by SEF neurons, we drew tuning curves of neuronal activity of an example neuron (Fig. 4f) and population activity (Fig. 4g) as a function of time throughout the trial. For each neuron, the spike count in each sliding window (duration, 200 ms; step size, 10 ms) was averaged across trials with the saccade target at the LD, L, LU, RU, R, or RD positions.

To draw the tuning curve of population activity (Fig. 4g), we first screened the neurons based on whether they had PD. For the preselection period and earlier period, the tuning curve of each neuron was reordered according to whether the PD of each neuron was in the left (LD, L, or LU) or right (RU, R, or RD) visual hemifield, so that the first three directions were those in the visual hemifield including the PD (PD-included) and the latter three directions were those that did not include the PD (PD-opposite). Then, the spike count was averaged across the neurons for each corresponding direction. For task periods after the choice targets were presented,

the tuning curve of each neuron was reordered by rotating it in six directions so that the PDs were aligned across the neurons. Values in the directions in which no data were obtained (Up and Down) were calculated using linear interpolation with the values in the adjacent directions. Population results were then obtained by calculating the mean across neurons.

### Classification of neurons

On the basis of the pattern of cell activity in the visually guided delayed saccade task, the neurons were classified into four functional types: visual, visuomovement, movement, or none. Each neuron was classified based on whether it exhibited visual-related activity, saccade-related activity, or both as judged with a Wilcoxon signed-rank test (α = 0.05 each). If a neuron exhibited a greater number of spikes after (0–300 ms) than before the (−300 ms to 0 ms) target onset, it was judged as showing visual-related activity. If a neuron exhibited a greater number of spikes around (−100 ms to 50 ms) than before (−350 ms to −200 ms) a saccade, it was judged as showing saccade-related activity. Neurons that exhibited visual-related activity, but not saccade-related activity, were classified as visual type (Fig. 6, left). Neurons that exhibited visual-related activity and saccade-related activity were classified as the visuomovement type (Fig. 6, center). Neurons that exhibited saccade-related activity, but not visual-related activity, were classified as movement type (Fig. 6, right).

### Electrical stimulation

Electrical stimulation experiments were conducted approximately 2–3 months after the recording experiments had been completed. In the electrical stimulation experiments, the duration of the symbolic cue period was shortened to 120 ms to increase the behavioral performance sensitivity by increasing the difficulty of the task and to minimize tissue damage by suppressing the duration for which stimulation was applied. Stimulation was applied during the symbolic cue period and the following 500-ms delay period via a tungsten epoxy-coated electrode (FHC; Bowdoin, ME; ~1.5 MΩ at 1 kHz) that was inserted into the SEF. Stimulation consisted of 206 trains of biphasic (0.3 ms per phase) cathodal-first rectangular constant-current pulses delivered at 333 Hz; 210–270 µA and 230 µA currents were used for monkeys K and A, respectively. The amplitude of the current was determined before each experimental session so that stimulation elicited saccades while the monkeys were freely viewing the blank screen but not while fixating on a visual stimulus. Indeed, we confirmed that saccades were not elicited when stimulation was applied during the task (Fig. 7a), which is explained by a previous finding that the threshold in the SEF when animals are attempting to hold fixation is 16 times greater than when fixation is not required[44]. Stimulation was applied in 50% of trials (stimulation trials) and not applied in the remaining 50% of trials (nonstimulation trials). Each block contained 16 stimulation trials (4 symbolic cues × 3 angles [−45°, 0°, and 45°] + 4 other angles) and 16 nonstimulation trials (4 symbolic cues × 3 angles [−45°, 0°, and 45°] + 4 other angles). Within each block, the order of these trial types was determined pseudorandomly, so the monkeys could not predict whether stimulation would be applied or not, the kind of symbolic cue, or the angle of the choice target. If a monkey failed to successfully complete a trial, the same trial type was not repeated within the same block.

### Reporting summary

Further information on research design is available in the Nature Portfolio Reporting Summary linked to this article.

## Data availability

The behavioral and neuronal data supporting the results reported in this paper are available on Zenodo under the identifier: https://doi.org/10.5281/zenodo.13282931[45].

## Code availability

The codes that generate the figures in this paper are available on Zenodo under the identifier: https://doi.org/10.5281/zenodo.13282931[45].

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

## Acknowledgements
We thank Dr. E. Hoshi for his invaluable comments on the manuscript and encouragement throughout this study. We also thank T. Ogata, N. Hashimoto, and D. Takahara for their technical assistance. Monkey A was supplied by the National BioResource Project (Japanese Monkeys) and

supported by the Ministry of Education, Culture, Sports, Science, and Technology of Japan. This work was supported by Grants-in-Aid for Scientific Research from the Japan Society for the Promotion of Science (KAKENHI 15K16016 and 18K07350 to OY and 18H05287 to YN).

## Author contributions

Conceptualization: O.Y. and Y.N. Methodology: O.Y. and Y.N. Software: O.Y. Formal analysis: O.Y. Investigation: O.Y. Resources: O.Y. and Y.N. Writing—original draft: O.Y. Writing—review and editing: O.Y. and Y.N.

## Competing interests

The authors declare no competing interests.
