## [Peer Review File · Communications Biology]

Reviewers' comments:

Reviewer #1 (Remarks to the Author):

In this manuscript, Yokoyama and Nishimura report findings of SEF recording and microstimulation experiments during a task in which symbolic cues indicate the hemifield in which an upcoming saccade target will appear. They report that a large number of SEF neurons encodes the potential target space, before the actual targets are presented. Microstimulation in the pretrial period caused a slight bias of spatial selection towards the contralateral hemisphere. The manuscript seems very mature. The rationale of the experiments is clearly described. It is easy to read, the analysis is clear and easy to follow. The interpretation of the findings is reasonable. I have no suggestions for improvements (a first for me in my career as a reviewer). I think this manuscript will make a valuable addition to the literature.

My only comment is with respect to the conceptualization of the process that is investigated. As I read the manuscript, I thought the 'visuospatial preselection under uncertainty' is basically top-down spatial attention. Do the authors disagree with that assessment. If yes, it might be worth pointing out why. If not, the authors might want to consider stating this more clearly. Showing that SEF is involved in spatial attention processing would be of great significance.

Reviewer #3 (Remarks to the Author):

The authors developed a novel task, which decouples the process of pre-selection of a potential target space from target selection. The authors claim that SEF neurons represent information about the potential target space during a pre-selection period, and demonstrate that electrical stimulation of SEF during the pre-selection period disrupts subsequent target selection.

Although the topic is of interest to the general neuroscience audience, I am of the opinion that the manuscript requires major revisions before being suitable for publication. The conclusions drawn in the manuscript have one major flaw. As reported in the manuscript, monkeys make micro-saccades toward the potential target space during the pre-selection period. Therefore, it is unclear whether the reported change in neural activity during the pre-selection period is related to pre-selection of a potential target space or to micro-saccadic eye movements. To convince the reader that changes in neural activity are not

related to micro-saccadic eye movements, the authors compare two regression models to predict spike count; one model only regressed by instructed target space, and one model regressed by instructed target space and eye position. The authors compare the resulting correlation coefficients and conclude that, because the resulting correlation coefficients are not significantly different, both models explain the data equally well.

However, there are several shortcomings with the approach: 1) It is unclear to me how correlation coefficients across time were calculated. Were these also calculated using a 200ms sliding window approach? 2) Correlation coefficients might be equal over time even though the two models do not explain the data equally well. The fit of two models is usually compared using F-statistics. Why has that not been done here? 3) Most importantly, the instructed target space/hemifield and eye position heavily correlate with each other. Therefore, it is not surprising that the two models explain the data equally well.

A better approach to test whether changes in neural activity during the pre-selection period are related to the pre-selection of a potential target space or micro-saccadic eye movements would be to compare neural activity across four different groups of trials: 1) Trials in which the rightward hemifield was cued and the monkey makes primarily rightward micro-saccades, 2) trials in which the rightward hemifield was cued and the monkey makes primarily leftward micro-saccades, 3) trials in which the leftward hemifield was cued and the monkey makes primarily leftward micro-saccades, and 4) trials in which the leftward hemifield was cued and the monkey makes primarily rightward micro-saccades. If the neural activity is comparable between the two groups of trials in which the right hemifield was cued and the two groups of trials in which the left hemifield was cued, changes in neural activity must be related to the pre-selection of a potential target rather than micro-saccadic eye movements.

If this analysis is not possible due to an insufficient number of trials in each group of trials, an alternative approach would be to compare changes in neural activity in the pre-selection period with changes in neural activity during the fixation period on the visually-guided saccade task. Supposedly, monkeys perform micro-saccades on both tasks and the authors would be able to compare changes in the neural activity between the two tasks on trials, in which the monkey makes primarily rightward micro-saccades, as well as changes in neural activity between the two tasks on trials, in which the monkey makes primarily leftward micro-saccades. If the neural activity is significantly different between the two tasks for both groups of trials, it is unlikely that changes in neural activity are primarily driven by micro-saccadic eye movements. However, there is one caveat with this approach. Micro-saccadic eye movements can differ significantly between tasks and this

approach does not account for these differences.

Page 6, line 114: Are these well-isolated single units or multi-units? Was spike sorting performed online or offline after the experiments? These details are later mentioned in the methods section. However, these details provide important context for understanding the results and should briefly be mentioned here.

Page 6, line 121: If success trials are trials in which the correct target was chosen, why do you not simply call them correct trials? Is there a difference between success and correct trials?

Page 7, line 145: Using 200-ms sliding windows results in a relatively low time resolution. For example, a sliding window that contains the first 10 to 190 ms of the delay phase also contains the last 190 to 10 ms of the symbolic cue phase. Therefore, it is unclear whether task-related activity observed in such a time window arose during the late symbolic cue phase or the early delay phase. It would be informative to show how the analysis compares when using a smaller sliding window, or when using alternative methods to determine the time course of task-related factors.

Page 10, line 210-213: Sentence is unclear, especially the part “indicating that presentation of a target in the PD enhanced neuronal activity, consistent with previous findings¹⁷, even in the presence of a distractor”. Please revise.

Page 12, line 251-258: Although the information is later provided in the methods section, it is unclear from the information provided in the results section how exactly neurons were classified as visual, visuomovement, or movement neurons. This information provides important context for understanding the results and should briefly be mentioned here.

Page 13, line 271: It is unclear how “spatial selection under uncertainty” differs from pre-selection of the target space.

Page 14, line 304-305: It is unclear whether the claim that “these results demonstrate that the activity of SEF neurons is causally involved in spatial selection for saccadic targets under uncertainty” holds. Electrical stimulation biases micro-saccadic eye movements in the pre-selection phase. It is unclear whether electrical stimulation biases neural activity related to pre-selection of a target space. Instead electrical stimulation might bias neural activity related to eye position, which then biases eye position and subsequently target selection. The authors even acknowledge this option themselves on page 15, line 330 –

page 16, line 335.

Page 15, line 328-329: Meaning of sentence is unclear. Please revise.

Page 16, line 335-337: Was the limited effect of SEF simulation mentioned prior in the manuscript? If not, it should first be mentioned in the results section before being discussed.

Page 16, line 349 - page 17, line 355: This is one possible explanation of the results but is presented as if it is the only possible explanation. Please revise to make it clear that this is the authors' interpretation of the results.

Page 17, line 370-371: Unclear, please expand on previous findings to provide additional context for the reader.

Page 24, line 528-530: The monkey still performed a lot of trials with a fixed association between symbolic cues and correct target position. How did you confirm that monkeys had not learned the association?

Page 25, line 570-573: Was there a minimum number of trials to include a neuron in the data analysis?

Page 26: The presented approach to determine preferred direction of a neuron would not capture neurons with wider tuning curves. However, neurons with a wider tuning curve still have preferred directions, albeit more than one. The authors either need to specify their definition of preferred direction or use an alternative approach that captures multiple preferred directions to account for neurons with wider tuning curves.

Page 29, line 712-715: Given that success rate in stimulation sessions dropped, it might be informative to additionally compare behavior between stimulation trials and control sessions.

Supplementary figure 1 keeps being referenced throughout the first half of the manuscript and appears to be a key figure. It should be a main figure rather than a supplementary figure. Figure 5 could instead be made a supplementary figure.

Thank you very much for your consideration of our manuscript, “**Preselection of potential target spaces based on partial information by the supplementary eye field**” (Tracking #: COMMSBIO-23-3829). We appreciate the careful review of our manuscript and would like to thank the two reviewers for their time spent writing detailed and helpful comments. We have incorporated all the suggestions, and our point-by-point responses to all their comments are attached below. Changes to the manuscript are shown in *red*. We believe that we completely address all the issues raised by the reviewers, as described below.

Reviewers' comments:

Reviewer #1 (Remarks to the Author):

In this manuscript, Yokoyama and Nishimura report findings of SEF recording and microstimulation experiments during a task in which symbolic cues indicate the hemifield in which an upcoming saccade target will appear. They report that a large number of SEF neurons encodes the potential target space, before the actual targets are presented. Microstimulation in the pretrial period caused a slight bias of spatial selection towards the contralateral hemisphere. The manuscript seems very mature. The rational of the experiments is clearly described. It is easy to read, the analysis is clear and easy to follow. The interpretation of the findings is reasonable. I have no suggestions for improvements (a first for me in my career as a reviewer). I think this manuscript will make a valuable addition to the literature.

My only comment is with respect to the conceptualization of the process that is investigated. As I read the manuscript, I thought the 'visuospatial preselection under uncertainty' is basically top-down spatial attention. Do the authors disagree with that assessment. If yes, it might be worth pointing out why. If not, the authors might want to consider stating this more clearly. Showing that SEF is involved in spatial attention processing would be of great significance.

We thank you for your positive evaluation and thoughtful comments. As you noted, we also suspected that ‘visuospatial selection under uncertainty’ might be accounted for by top-down spatial attention. However, as we did not conduct any direct tests that dissociated visuospatial attention and saccade preparation in the present study, we are not able to conclude that the modulation of SEF neuronal activity during the preselection period was accounted for by visuospatial attention. We *proposed* that the present findings are related to visuospatial attention in the Discussion (in the third subsection of the Discussion “Possible roles of the spatial information encoded by SEF neurons”). We revised this subsection to make it clearer as follows (page 19, lines 411–424):

One possibility is that spatial encoding by SEF neurons is related to visuospatial attention. Mechanistically, as mentioned in the previous subsection, by increasing the discharge rate prior to the target's appearance, the visual response to a target candidate appearing in the neuron's PD reaches a certain firing rate **more quickly** than the visual response to another target candidate appearing outside the preselected space (Fig. 4). This facilitates the detection and selection of a target appearing in the preselected space but not outside it. This **mechanism** seems to be **similar to** that **underlying** visual attention in the visual brain areas³¹. **Additionally**, we **observed** that the gaze was slightly displaced toward the selected visual hemifield during the preselection period (Fig. 5b) and SEF stimulation (Fig. 7c). **This** finding is similar to previous findings that microsaccades occur in the direction of attention during covert attention^{19,20}. Finally, mainly visual-type neurons represented the potential target space during the preselection period (Fig. 6). Collectively, the present results suggest that the role of SEF activity **in this** task seems **to be** more **related** to visuospatial attention. Indeed, previous studies of single units in monkeys¹⁷ and human neuroimaging studies³² have implicated the SEF in spatial attention when the target position is certain.

Reviewer #3 (Remarks to the Author):

The authors developed a novel task, which decouples the process of pre-selection of a potential target space from target selection. The authors claim that SEF neurons represent information about the potential target space during a pre-selection period, and demonstrate that electrical stimulation of SEF during the pre-selection period disrupts subsequent target selection.

Although the topic is of interest to the general neuroscience audience, I am of the opinion that the manuscript requires major revisions before being suitable for publication. The conclusions drawn in the manuscript have one major flaw. As reported in the manuscript, monkeys make micro-saccades toward the potential target space during the pre-selection period. Therefore, it is unclear whether the reported change in neural activity during the pre-selection period is related to pre-selection of a potential target space or to micro-saccadic eye movements. To convince the reader that changes in neural activity are not related to micro-saccadic eye movements, the authors compare two regression models to predict spike count; one model only regressed by instructed target space, and one model regressed by instructed target space and eye position. The authors compare the resulting correlation coefficients and conclude that, because the resulting correlation coefficients are not significantly different, both models explain the data equally well.

However, there are several shortcomings with the approach: 1) It is unclear to me how correlation coefficients across time were calculated. Were these also calculated using a 200ms sliding window approach? 2) Correlation coefficients might be equal over time even though the two models do not explain the data equally well. The fit of two models is usually compared using F-statistics. Why has that not been done here? 3) Most importantly, the instructed target space/hemifield and eye position heavily correlate with each other. Therefore, it is not surprising that the two models explain the data equally well.

A better approach to test whether changes in neural activity during the pre-selection period are related to the pre-selection of a potential target space or micro-saccadic eye movements would be to compare neural activity across four different groups of trials: 1) Trials in which the rightward hemifield was cued and the monkey makes primarily rightward micro-saccades, 2) trials in which the rightward hemifield was cued and the monkey makes primarily leftward micro-saccades, 3) trials in which the leftward hemifield was cued and the monkey makes primarily leftward micro-saccades, and 4) trials in which the leftward hemifield was cued and the monkey makes primarily rightward microsaccades. If the neural activity is comparable between the two groups of trials in which the right hemifield was cued and the two groups of trials in which the left hemifield was cued, changes in neural activity must be related to the pre-selection of a potential target rather than micro-saccadic eye movements.

If this analysis is not possible due to an insufficient number of trials in each group of trials, an alternative approach would be to compare changes in neural activity in the pre-selection period with changes in neural activity during the fixation period on the visually-guided saccade task. Supposedly, monkeys perform micro-saccades on both tasks and the authors would be able to compare changes in the neural activity between the two tasks on trials, in which the monkey makes primarily rightward micro-saccades, as well as changes in neural activity between the two tasks on trials, in which the monkey makes primarily leftward micro-saccades. If the neural activity is significantly different between the two tasks for both groups of trials, it is unlikely that changes in neural activity are primarily driven by micro-saccadic eye movements. However, there is one caveat with this approach. Micro-saccadic eye movements can differ significantly between tasks and this approach does not account for these differences.

We thank you for your thoughtful comments and practical suggestions for better analyses on the relationship between neuronal activity and eye movements. We have reanalyzed the eye movement and neuronal activity data according to your advice, as described below. We believe that these revisions have significantly improved the quality of our manuscript.

As you suggested, we first reanalyzed trial-by-trial data on eye movement during the preselection period to determine whether there were trials in which the eye moved to the hemifield opposite to the instructed hemifield. We found that in approximately 10–40% of the trials in each experimental session, the eye moved to the hemifield opposite to the instructed hemifield rather than to the instructed hemifield. In an example session of Monkey P, in the left-instructed trials, the eye moved to the left in 116 trials and to the right in 44 trials, while in the right-instructed trials, the eye shifted to the left in 67 trials and to the right in 93 trials (left side of the figure below), resulting in four groups of trials in terms of the combination of the instructed hemifield and the direction of eye position shift. Next, we compared the firing rate of each neuron between trials in which the eye shifted in the opposite direction and trials in which the eye shifted to the instructed hemifield. The result of an example neuron is shown in the right side of the figure below. This neuron exhibited a clear modulation in its firing rate depending on the instructed hemifield but much less on the direction of the eye position shift. In response to the symbolic cue instructing the left hemifield, the firing rate increased to more than 30 spikes/s, whereas the firing rate decreased to almost zero in response to the symbolic cue instructing the right hemifield. In contrast, the same neuron showed less modulation in its firing rate depending on the direction of eye position shift (brown versus orange and purple versus light purple lines). To test this statistically, we conducted two-way ANOVAs using the instruction (left/right) and the direction of eye position shift (left/right) as factors in 10-ms sliding windows with a 10-ms step size for the inverse interspike interval (iISI) (see below) of the activity of each neuron to examine whether the neuronal activity was modulated depending on the instructions, the direction of eye position shift, or both. The results of this statistical test for the example neuron are shown on the top of the plot by a blue line indicating the timepoints with significant effects of the instructions ($p < 0.01$ in ≥ 3 consecutive bins). The statistical test also revealed that the activity of this neuron did not differ depending on the direction of the eye position shift throughout the preselection period.

For all of the recorded neurons, we calculated the proportion of neurons with a significant effect of the instructions or the direction of the eye position shift in each of the sliding windows (figure below).

The results showed that the proportion of neurons in which the firing rate was modulated by the instructions rapidly increased in response to the symbolic cue up to approximately 20%, whereas the proportion of neurons in which the firing rate differed according to the direction of eye position shift was <5% throughout the phase. These results demonstrate that a majority of the observed changes in neuronal activity during the preselection period are much more closely related to preselection of a potential target space than to eye movements. We have added these new results to the manuscript, which has been reorganized to present the results of the analyses of eye position shift and the results of the analysis of the effects of eye position on neuronal activity together (Fig. 5 and pages 11–12, lines 232–253). We have also added a sentence describing this important result to the first paragraph of the Discussion in the main text to clearly state that the activity changes in SEF neurons were mostly related to the instructed hemifield but not to eye position bias, as follows (page 16, lines 340–341): **These activity changes were scarcely accounted for by eye position.**

Page 6, line 114: Are these well-isolated single units or multi-units? Was spike sorting performed online or offline after the experiments? These details are later mentioned in the methods section. However, these details provide important context for understanding the results and should briefly be mentioned here.

We have revised the manuscript to add a description about our inclusion of only well-isolated single units in our analyses as follows (page 6, lines 106–108):

We **analyzed** the activity of a total of 201 neurons, **which were well isolated offline after the experiments**, across the left SEFs of two monkeys (monkeys A and P, 110 and 91 neurons, respectively; Fig. 1d) while they performed the saccadic choice task (Fig. 1a).

Page 6, line 121: If success trials are trials in which the correct target was chosen, why do you not simply call them correct trials? Is there a difference between success and correct trials?

In the present study, we analyzed only the trials in which the correct target was chosen and the criteria for the eye position were fulfilled. We believe that the control of eye position and the correct target choice are both important and necessary for the animals to successfully complete the task. For this reason, we would like to use the term “success trials” in the manuscript. We have revised the manuscript to clarify this point (page 6, lines 115–116):

In the following analyses, only the success trials (trials in which the correct target was chosen **and the criteria for eye position were fulfilled**) were included.

Page 7, line 145: Using 200-ms sliding windows results in a relatively low time resolution. For example, a sliding window that contains the first 10 to 190 ms of the delay phase also contains the last 190 to 10 ms of the symbolic cue phase. Therefore, it is unclear whether task-related activity observed in such a time window arose during the late symbolic cue phase or the early delay phase. It would be informative to show how the analysis compares when using a smaller sliding window, or when using alternative methods to determine the time course of task-related factors.

Thank you for your thoughtful comments. As suggested, we performed sliding window analysis using shorter sliding windows than those used in the previously submitted version of the manuscript. Due to the improved temporal resolution, the changes in neuronal activity in temporal relation to the task phase became clearer, as described in detail below.

For Figure 3, to obtain a better temporal resolution for neuronal activity, we performed the inverse interspike interval analysis (1/ISI) (Coe, 2002). In this analysis, for any two consecutive spikes, the length of the interspike interval was assigned to the time period of the first spike and to every period between the two spikes. The spike times were measured at a 1-millisecond resolution so that interval functions with 1-millisecond bins were created. Then, we took the inverse of each ISI (1/ISI) to create spike-frequency functions, resulting in each trial having an activity score at every 1 millisecond interval with values of >0 and ≤ 1 kHz. Finally, we performed a statistical analysis (a set of two-way ANOVAs) of the inverse ISI using sliding windows of 10-ms bins with a 10-ms step size to examine which task-related factors were represented by the neuronal activity. We compared the results obtained by using the new analysis (mean inverse ISI in a window size of 10 ms with a step size of 10 ms) with the results obtained by using the previous version of the analysis (mean spike count in a window size of 200 ms with a step size of 10 ms). As shown in the figure below, the overall temporal pattern of the represented factors across all the recorded neurons looked qualitatively similar between the previous

(left, Figure 3(c) in the previously submitted manuscript) and the new (right) results.

Similarly, the temporal pattern of the proportion of the neurons that represented each factor was similar, although the proportions were somewhat smaller (approximately 3%) in the new results than in the previous results (left).

The new results using a smaller window size clearly showed that the proportion of neurons that represented the potential target space increased until the end of the symbolic cue phase, temporarily decreased in the early delay phase, and then increased again in the middle to late delay phase.

Additionally, for the analyses based on the preferred direction (PD) of individual neurons (Figure 4(d)–(f) in the previous manuscript), we reanalyzed the same data using a smaller time window (window size: 100 ms, step size: 100 ms) than that used in the previous version (window size: 200 ms, step size: 50 ms). The new sliding window and step size were used to examine the effect of conditions (PD-included and PD-opposite) with clear delineation of the symbolic cue phase and the delay phase by avoiding the placement of individual analysis windows on the temporal border of these phases. For

the analysis of the number of neurons that encoded the instructed visual hemifield when it included their PDs (Figure 4(d)), the resulting patterns of the new results (right side of the figure below) were similar to those of the previous analysis (left side of the figure below), but the new results clearly showed statistically significant differences in the middle symbolic cue phase and the early and late delay phases but not in the late symbolic cue phase.

Additionally, for the analysis of the area under the ROC curve (Figure 4(e) in the previous manuscript), we reanalyzed the same data using a smaller time window (window size: 100 ms, step size: 100 ms) than that used in the previous analysis (window size: 200 ms, step size: 50 ms). As shown in the figures below, the new results (right) showed a temporal pattern similar to that of the previous results (left). The statistical analysis of the new results clearly showed that there were significant differences between the conditions during the middle symbolic cue phase and the middle to late delay phase but not in the late symbolic phase.

Finally, for the regression analysis (Figure 4(f) in the previous manuscript), we reanalyzed the same data using the same windows used above (window size: 100 ms, step size: 100 ms). The new result (right side of the figure below) showed a temporal pattern similar to that of the previous analysis (left side of the figure below) and clearly showed that there were statistically significant differences in the

middle symbolic cue phase and the middle to late delay phase but not in the late symbolic or early delay phase.

Based on the results described above, we have replaced the previous figures with those obtained in the reanalyses using smaller time windows in the revised version of our manuscript. We also revised the related part of the main text (in the Results and Methods sections [pages 7–9]), as shown in red in the revised version of our manuscript.

Page 10, line 210-213: Sentence is unclear, especially the part “indicating that presentation of a target in the PD enhanced neuronal activity, consistent with previous findings¹⁷, even in the presence of a distractor”. Please revise.

We have revised the text as follows (page 10, lines 207–212):

For specified target location, we confirmed that the SEF neurons exhibited greater activity when the chosen target was in the same direction as their PD than when it was not during the target determination period (Supplementary Fig. 2). This result is similar to a previous finding¹⁷ that SEF neurons show greater activity when a single stimulus is presented in the neuron’s PD, but our results extend this finding and note that SEF neurons also exhibit greater activity when a stimulus is presented in the PD even when another stimulus is presented outside the PD.

Page 12, line 251-258: Although the information is later provided in the methods section, it is unclear from the information provided in the results section how exactly neurons were classified as visual, visuomovement, or movement neurons. This information provides important context for understanding the results and should briefly be mentioned here.

We have added a description of how we classified the neurons into visual, visuomovement, or movement types as follows (page 13, lines 270–284):

On the basis of the activity pattern observed in the visually guided delayed saccade task (Fig. 1b), SEF neurons can be characterized as having a visual and/or movement component¹⁸. We identified three types of neurons, i.e., *visual*, *visuomovement*, and *movement*, and examined which type encoded the potential target space (Fig. 6). Neurons that increased their activity in response to visual stimulation ($\text{spike number}_{[0-300 \text{ ms after target onset}]} > \text{spike number}_{[0-300 \text{ ms before target onset}]}$, $\alpha = 0.05$ by Wilcoxon signed-rank test), but not around saccades ($\text{spike number}_{[-100-50 \text{ ms around saccade onset}]} = \text{spike number}_{[350-200 \text{ ms before saccade onset}]}$, $\alpha = 0.05$ by Wilcoxon signed-rank test), were classified as *visual* type ($n = 34/201$ [17%]; Fig. 6a, left). Neurons that increased their activity in response to visual stimulation ($\text{spike count}_{[0-300 \text{ ms after target onset}]} > \text{spike count}_{[0-300 \text{ ms before target onset}]}$, $\alpha = 0.05$ by Wilcoxon signed-rank test) and around saccades ($\text{spike count}_{[-100-50 \text{ ms around saccade onset}]} > \text{spike count}_{[350-200 \text{ ms before saccade onset}]}$, $\alpha = 0.05$ by Wilcoxon signed-rank test) were classified as *visuomovement* type ($n = 4/201$ [2%]; Fig. 6a, center). Neurons that increased their activity around saccades ($\text{spike count}_{[-100-50 \text{ ms around saccade onset}]} > \text{spike count}_{[350-200 \text{ ms before saccade onset}]}$, $\alpha = 0.05$ by Wilcoxon signed-rank test), but not in response to visual stimulation ($\text{spike count}_{[0-300 \text{ ms after target onset}]} = \text{spike count}_{[0-300 \text{ ms before target onset}]}$, $\alpha = 0.05$ by Wilcoxon signed-rank test), were classified as *movement* type ($n = 17/201$ [8%]; Fig. 6a, right).

Page 13, line 271: It is unclear how “spatial selection under uncertainty” differs from preselection of the target space.

We have revised this part of the manuscript as follows (page 14, lines 297–300):

...however, it remained unclear whether the SEF contributes to the preselection of potential target space. Therefore, we tested whether the activity of SEF neurons was causally involved in preselection of potential target space by performing electrical stimulation experiments in two monkeys (monkeys A and K, 6 and 7 sessions, respectively).

Page 14, line 304-305: It is unclear whether the claim that “these results demonstrate that the activity of SEF neurons is causally involved in spatial selection for saccadic targets under uncertainty” holds. Electrical stimulation biases micro-saccadic eye movements in the preselection phase. It is unclear whether electrical stimulation biases neural activity related to preselection of a target space. Instead electrical stimulation might bias neural activity related to eye position, which then biases eye position and subsequently target selection. The authors even acknowledge this option themselves on page 15, line 330 – page 16, line 335.

Thank you for the thoughtful comment. Electrical stimulation influences the activity of neurons within several millimeters around the electrode tip regardless of their anatomical connections and functions. Thus, as you pointed out, in the present study, we were unable to differentiate between the possibility that electrical stimulation biased neuronal activity related to preselection of the target space and the possibility that electrical stimulation biased neuronal activity related to eye movement. However, we believe that we can argue that “these results demonstrate that the activity of SEF neurons is *causally* involved in spatial selection”, regardless of whether the effect was direct or indirect via eye position shift. Generally, in the literature (ex. <https://doi.org/10.1073/pnas.2221641120>), the term “causal” is used when a cause and an effect are identified even if the mediating mechanisms have not been fully clarified. As you noticed, we acknowledge this point and discuss it in the Discussion section. As shown in Figure 5c–f, the correspondence between the selected hemifield and the eye position bias was weak, suggesting that eye position bias might not mediate target selection; rather, both eye position bias and target selection seem to be influenced by the activity of SEF neurons. We have also added this discussion point to the Discussion as follows (page 17, lines 364–368):

Since the correspondence between the selected hemifield and the direction of the eye position shift was weak (Fig. 5c, d) and the population of SEF neurons representing eye position was much smaller than that of SEF neurons representing the instructed visual hemifield (Fig. 5e), the effect of electrical stimulation on target selection is unlikely to be mediated by neurons involved in eye movements.

Page 15, line 328-329: Meaning of sentence is unclear. Please revise.

We have revised the sentence to clearly show that our results showing the contraversive effect of SEF stimulation in selecting an expanse of space when the target position is uncertain expands upon the previous stimulation and lesion studies^{21–25} that reported the contralateral bias of SEF functions, as follows (page 16, lines 357–358):

Our results add to these studies the **contralateral bias** of the SEF in selecting an expanse of space **when the target position is uncertain**.

Page 16, line 335-337: Was the limited effect of SEF simulation mentioned prior in the manuscript? If not, it should first be mentioned in the results section before being discussed.

In that part of the Discussion section, we intended to use the term “limited effect” to indicate a “small effect” since SEF stimulation in ipsilateral trials induced significantly more erroneous choices of the

contralateral option (5% increase on average) but did not completely suppress correct choices. As noted, we have revised the Results section to clarify that SEF stimulation induced a significant but limited effect as follows (page 15, lines 323–326):

This decrease in the correct choice rate in the ipsilateral trials with SEF stimulation (-0.05 , mean \pm SEM = 0.86 ± 0.019 in stimulation trials compared to 0.91 ± 0.021 in nonstimulation trials) was **small, but statistically significant** (Fig. 7f, left; $n = 16$, paired t test, $t(15) = 3.65$, $p = 0.0024$).

Page 16, line 349 - page 17, line 355: This is one possible explanation of the results but is presented as if it is the only possible explanation. Please revise to make it clear that this is the authors' interpretation of the results.

We have revised this part of the Discussion section to clarify what is demonstrated by the present results and what our interpretation is (pages 17–18, lines 382–389):

When preselecting a region of space beyond that of the PD of each neuron, a subset of neurons that have various PDs in the potential target space exhibited an increase in their firing rate; this process could lead to the selection of a region of space by superposition of the neurons' PDs. The greater firing rate of these neurons before target appearance could then result in a greater firing rate in response to targets appearing in the preselected target space (a collection of PDs) compared to the rate elicited by targets appearing outside the potential target space (Fig. 4). We propose that this scenario is a plausible neurophysiological mechanism implemented by the SEF that underlies spatial preselection based on partial information, even before any target candidates appear.

Page 17, line 370-371: Unclear, please expand on previous findings to provide additional context for the reader.

We have revised this part of the Discussion section as follows (page 18, lines 403–406):

Previous behavioral studies have shown that the efficiency of attentional processing declines gradually, but not sharply, around the attended space^{29,30}. The inevitable consequence of the superposition of broad tuning curves of a population of neurons may be one of the reasons why the boundary between the selected and unselected space is not sharply demarcated.

Page 24, line 528-530: The monkey still performed a lot of trials with a fixed association between symbolic cues and correct target position. How did you confirm that monkeys had not learned

the association?

Thank you for your insightful comments. We reconsidered this point. We agree that the monkeys performed the task based on the associations between symbolic cues and correct target positions. We hope that monkeys expectantly select (i.e., preselect) either the left or right visual hemifield in response to the symbolic cues. Several observations suggest that it is implausible that the monkeys relied only on the separate associations. As described in the first subsection of the Results section, after the training, the monkeys were able to make correct choices with high accuracy (>93%) in all the trial types with choice targets presented at different angles (a total of 18 different angles [$\pm 45^\circ$, $\pm 40^\circ$, $\pm 35^\circ$, $\pm 30^\circ$, $\pm 25^\circ$, $\pm 20^\circ$, $\pm 15^\circ$, $\pm 10^\circ$, $\pm 5^\circ$]). Additionally, we obtained results showing the bias of eye position and the modulation of neuronal activity during the preselection period, which are consistent with the preselection of either visual hemifield, suggesting that the process of preselection of either hemifield started even during the preselection period. We believe that these results suggest that we succeeded in attempting to have the monkeys preselect either visual hemifield during the preselection period rather than independently learn each of the associations between symbolic cues and correct target positions. Based on these considerations, we have revised the sentence as follows (page 25, lines 562–564):

These additional trials were used to facilitate preselection of either the left or right visual hemifield even when the target position was still uncertain.

Page 25, line 570-573: Was there a minimum number of trials to include a neuron in the data analysis?

We have added the following description about the range and average number of trials per neuron included in our analyses (page 26, lines 608–609):

The number of trials per neuron included in the analyses ranged from 200–667 trials (mean = 452 trials).

Page 26: The presented approach to determine preferred direction of a neuron would not capture neurons with wider tuning curves. However, neurons with a wider tuning curve still have preferred directions, albeit more than one. The authors either need to specify their definition of preferred direction or use an alternative approach that captures multiple preferred directions to account for neurons with wider tuning curves.

As you noted, most of the SEF neurons had broad tuning curves; that is, the neurons showed an

increase in firing rate in response to the appearance of a target not only in one particular direction but also in adjacent directions. In the present study, to clarify that the activity of SEF neurons was modulated depending on the relationship between the direction selectivity of the neurons and the space to be preselected, we defined PD as the direction in which a target elicited the largest significant increase in the firing rate of the neuron. We revised the portion of the manuscript as follows (page 25, lines 573–576):

A conventional visually guided delayed saccade task (Fig. 1b) was used to determine the basic response properties of individual neurons, such as their preferred direction (PD, see the definition described in the subsection “Relation between PD and representation of the instructed hemifield” below)

Page 29, line 712-715: Given that success rate in stimulation sessions dropped, it might be informative to additionally compare behavior between stimulation trials and control sessions.

According to your comments, we have reanalyzed the behavioral data and have added a figure (Fig. 7e) comparing the success rate between the sessions in which stimulation was applied and the sessions in which stimulation was not applied, which were performed around the same time period. As you suggested, the results showed that the rate of success choice was significantly lower in stimulation sessions than in nonstimulation sessions, indicating that electrical stimulation of the SEF disrupted overall choice behavior that depended on the preselection of space. This result was also described in the main text (Page 15, lines 319–322). Additionally, we replaced the figure (Fig. 7f) comparing success choice rate between stimulation trials and nonstimulation trials with a figure including all the sessions ($n = 16$) conducted, which resulted in a graph that is qualitatively similar to the previous version that included only the sessions ($n = 13$) in which success choice rates in both the nonstimulation ipsilateral and contralateral trials were greater than 0.75. The figure showing the eye position during the preselection period in the electrical stimulation experiments (Fig. 7c, d) was also replaced accordingly to include all the sessions ($n = 16$). The statistical value described in the main text was changed accordingly (Page 15).

Supplementary figure 1 keeps being referenced throughout the first half of the manuscript and appears to be a key figure. It should be a main figure rather than a supplementary figure. Figure 5 could instead be made a supplementary figure.

Thank you very much for this valuable comment. According to your comment, we have relocated the

figure depicting the visually guided delayed saccade task (Supplementary figure 1a, b) from the supplementary material to the main figure (Fig. 1b), which is presented together with the figure depicting the saccadic choice task. We have also added a new figure illustrating the order in which these tasks were conducted in each recording session, as shown in Fig. 1c. In addition, we have included the figures showing the activity of an example neuron in the visually guided saccade task in Fig. 2c, d in the same figure showing the activity of the same neuron in the saccadic choice task (Fig. 2a, b). Furthermore, we have included the figures showing population activity in the visually guided saccade task (Fig. 3b) with population activity in the saccadic choice task (Fig. 3a).

Fig. 5 in the previous manuscript shows the activity changes throughout the trial, spanning both the preselection period and the target determination period; as you pointed out, Fig. 5 is referred to in the manuscript only once. However, we believe that Fig. 5 illustrates the coherent relationship between the preference of neurons for the visual hemifield in the preselection period and the preference for the direction in the target determination period, which is one of the important findings of the present study. Therefore, we would like to keep the figure as a main figure. We have included the figure in Fig. 4 (Fig. 4f and 4g), which shows that SEF neurons encode the instructed visual hemifield if it includes their PDs.

REVIEWERS' COMMENTS:

Reviewer #1 (Remarks to the Author):

This manuscript is in excellent shape. I already was very satisfied with it and the authors have clearly responded to my (minor) concern. I think this manuscript is ready to be published.

Reviewer #3 (Remarks to the Author):

Yokoyama and Nishimura adequately addressed all comments and concerns raised in my previous review. The added statistical analyses support the interpretation of the findings by the authors and strengthen the claim that SEF neurons encode the potential target spaces during a pre-selection period. Given the added statistical analyses, the interpretation of the findings by the authors is reasonable. I have no further suggestions for improvements and recommend the manuscript to be published in its current form. I believe that the manuscript will influence the interpretation of previous and future findings in the neuroscience field.